# Non-hydrolyzable acetyllysine analogs to study protein acetylation in vitro and in cells

Simon Maria Kienle [1,2,4] ✉, Matthias Sigg [1,4], Tobias Schneider[2,3], Katrin Stuber [1,2,3], Jan Lehmann[1,2], Jasmin Jansen[1,2], Florian Stengel [1,2], Andreas Marx [2,3], Michael Kovermann [2,3] & Martin Scheffner [1,2] ✉

Lysine acetylation plays a prominent regulatory role in eukaryotic cells. Yet, determining the functional consequences of acetylation for a given protein represents a considerable challenge. For instance, lysine residues are subject to various posttranslational modifications, rendering interpretation of mutational studies difficult. The genetic code expansion technology enables site-specific incorporation of acetyllysine (AcK) into proteins, but the applicability of AcK is limited, as within cells, the acetyl group is removed by deacetylases. Here, we show that site-specific incorporation of the non-hydrolyzable AcK analog ketolysine (KeK) into ubiquitin closely resembles the structural and functional effects of AcK incorporation. Furthermore, AcK and KeK can be efficiently incorporated into the tumor suppressor p53 in cells. However, whereas AcK becomes deacetylated, KeK remains stable. Accordingly, incorporation of KeK, but not AcK, affects p53-mediated transcription. Thus, we propose that KeK is a well-suited AcK surrogate for studying acetylation of a given protein in cells.

The functions and properties of eukaryotic proteins are controlled by various posttranslational modifications (PTMs) that in most cases are reversible[1]. This allows a cell to rapidly adapt its proteome during time-regulated processes such as the cell cycle or differentiation and to respond to environmental changes. The continuous advancement and refinement of mass spectrometry (MS)-based approaches has been instrumental in identifying and cataloging different PTMs, e.g., phosphorylation, acetylation, and modification of proteins by ubiquitin (Ub) and Ub-like proteins, for individual proteins as well as on a proteome-wide scale[2,3]. Yet, while such studies provide invaluable insights into the plasticity of proteomes, how a given PTM affects the properties of a protein of interest (POI) remains unclear in many cases[4].

A general limitation in studying the effect of PTMs on the biochemical and structural properties of a POI is the generation of sufficient amounts of respectively modified proteins[5]. While in principle this can be achieved by incubation of the POI with the respective modifying enzymes, the efficiency of such reactions is often limited and/or not position-specific (i.e., residues in addition to, or other than the one of interest are modified), resulting in heterogeneous populations of the POI. Moreover, evaluation or validation of the results obtained in vitro may be even more challenging in cells. Common approaches to study the effect of a PTM in cells include the use of POI variants, in which the respective residue has been mutated (e.g., in case of lysine acetylation, replacement of lysine by arginine or glutamine), and modulation of the activity of the respective modifying enzyme[6]. Such approaches have obvious limitations. Targeting the modifying enzyme does in most, if not all cases not only affect the modification status of the POI, but also that of other proteins. Concerning replacement strategies, a main consequence of lysine acetylation, for instance, is neutralization of the positive charge of the side chain of lysine[7]. To provide evidence for the importance of a positive charge at the respective position of a POI, lysine is commonly replaced by arginine. However, lysine is not only subject to acetylation, but also to

[1]Department of Biology, University of Konstanz, Konstanz, Germany. [2]Konstanz Research School Chemical Biology, University of Konstanz, Konstanz, Germany. [3]Department of Chemistry, University of Konstanz, Konstanz, Germany. [4]These authors contributed equally: Simon Maria Kienle, Matthias Sigg. ✉e-mail: simon.kienle@uni-konstanz.de; martin.scheffner@uni-konstanz.de

other PTMs including modification by methyl groups, Ub, and Ub-like proteins[8]. Thus, it is difficult, or even not possible to distinguish if a respective effect is due to interference with acetylation or, for example, with ubiquitination. As surrogate for acetylated lysine (acetyllysine or briefly, AcK), glutamine is generally employed, which mimics elimination of the positive charge of the lysine side chain. However, at least in some cases, the functional consequences of lysine acetylation are not solely explained by charge neutralization[9–11]. Attachment of an acetyl group results in extension of the length of the side chain, which depending on the position in a POI, may induce structural changes, an effect that cannot be mimicked by the side chain of glutamine (see Fig. 1a), as we have for instance shown using Ub as model system[11].

The genetic code expansion technology represents an attractive possibility to incorporate AcK at the desired position of a POI, e.g., by the so-called amber codon suppression (ACS) method[5,12–14]. Yet, in applications in eukaryotic cells the use of K-to-Q variants may seem to be superior to ACS-mediated incorporation of AcK, as the latter is technically more complex and more importantly, the acetyl group can be readily removed by the action of deacetylases. To overcome this potential limitation, non-hydrolyzable AcK analogs, such as trifluoroacetyllysine (TFAcK)[15] and 2-amino-8-oxononanoic acid, or briefly ketolysine (KeK)[16], should prove helpful (Fig. 1a).

In this work, we use Ub[11] and the tumor suppressor p53[17–19] as model proteins in vitro and in cells, respectively, to prove the above proposition. The results we obtain with Ub in vitro clearly indicate that unlike Q, TFAcK and KeK represent highly suitable AcK surrogates. Furthermore, we provide evidence that upon incorporation into p53, AcK, and in part TFAcK, are efficiently deacetylated in cells, while KeK is refractory to deacetylation. Notably, this difference in deacetylation susceptibility correlates with the transcriptional activity of p53 variants harboring either AcK, TFAcK or KeK. In conclusion, our data indicate that KeK is a well-suited AcK surrogate for studying protein acetylation in cells.

## Results

### Incorporation of AcK, TFAcK, or KeK has similar effects on Ub structure and function

Besides its well-established role as a covalent modifier of itself and other proteins, Ub is also subject to PTMs including acetylation[20–22]. By employing the ACS technology, we have recently shown that acetylation of K11 affects the structure, and in consequence functional aspects of Ub[11]. Notably, these effects of acetylation are not caused solely by charge neutralization, but also involve steric effects presumably imposed by the length of the side chain (Fig. 1a). Indeed, replacement of K11 by the presumed AcK surrogate Q shows different effects on the structure of Ub, and the respective Ub variant is likewise functionally different, i.e., it is more efficiently used by the E3 ligase HDM2 for autoubiquitination than K11-acetylated Ub[11] (see also Fig. 1b). Together with results from other groups[9,10], our studies with Ub therefore indicate that data obtained with protein variants, in which lysine residues were replaced by glutamine, have to be interpreted with caution, especially in the absence of structural data.

AcK is likely subjected to deacetylation in cells. Therefore, we resorted to the AcK analogs TFAcK[23] and KeK[16], as their side chains are similar in length to AcK (Fig. 1a), but are either less (TFAcK)[15] or not (KeK) susceptible to deacetylation (see below). TFAcK or KeK were incorporated into Ub at position 11 (Ub 11X, where X represents the respective amino acid) by ACS in bacteria (Supplementary Fig. 1), and the respective Ub variants were first tested for their ability to be used by HDM2 in autoubiquitination experiments (Fig. 1b). In agreement with our previous results[11], Ub 11AcK was less efficiently used by HDM2 than unmodified Ub and Ub 11R (as judged by the levels of free Ub at 60 min), while Ub 11Q was more efficiently used than Ub 11AcK (Fig. 1b). Importantly, the efficiency of HDM2 autoubiquitination in the presence of Ub 11KeK or Ub 11TFAcK was similar to that in the presence of Ub 11AcK (Fig. 1b).

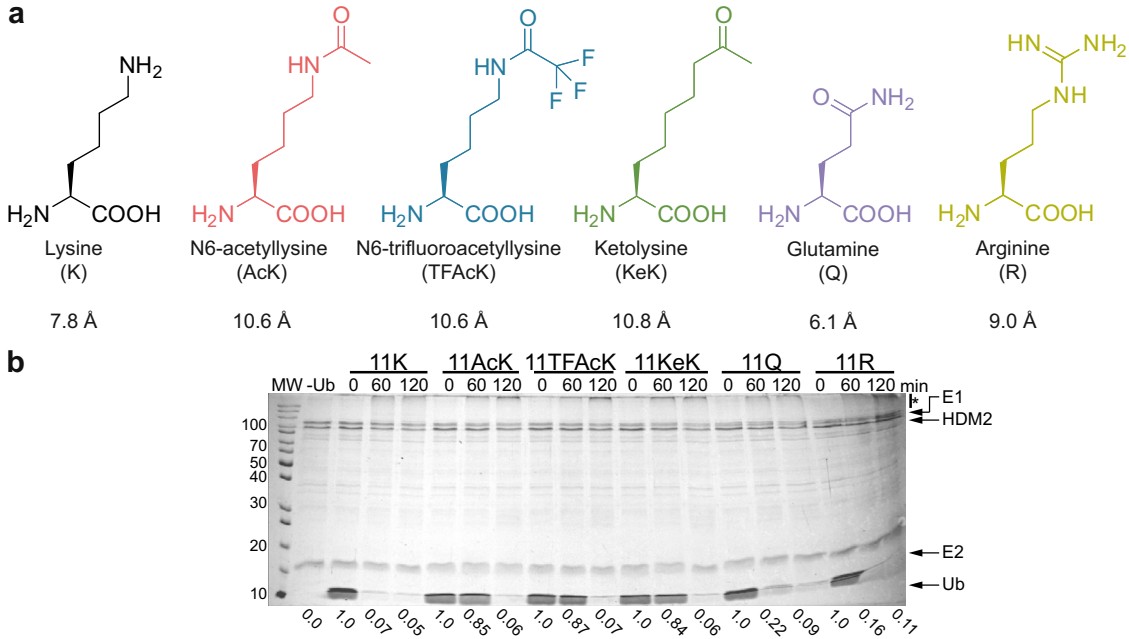

**Fig. 1 | The length of the side chain at position 11 of Ub affects HDM2 autoubiquitination. a** Comparison of the length (Å) of the side chain of amino acids incorporated at position 11 of Ub. **b** HDM2 autoubiquitination assay with Ub variants, in which K11 was replaced by AcK, TFAcK, KeK, Q or R, indicate that besides charge, the length of the side chain at position 11 is the main determinant for the performance of Ub variants in HDM2 autoubiquitination. Autoubiquitination reactions were started by addition of the respective Ub variant. Reactions were stopped at the times indicated. Reaction in the absence of Ub (-Ub) was stopped at 120 min. All reactions were analyzed by SDS-PAGE followed by Coomassie blue staining. Relative levels of free Ub variants were quantified with levels at 0 min set to 1. Running positions of molecular mass markers (MW, kDa), HDM2, UBA1 (E1), UbcH5b (E2), free Ub (Ub), and autoubiquitinated forms of HDM2 (*) are indicated. Quantitative incorporation of the respective amino acids and the homogeneity of the purified Ub variants were confirmed by ESI-MS analysis (Supplementary Fig. 1). The experiment shown is representative of three independent experiments. Source data are provided as a Source Data file.

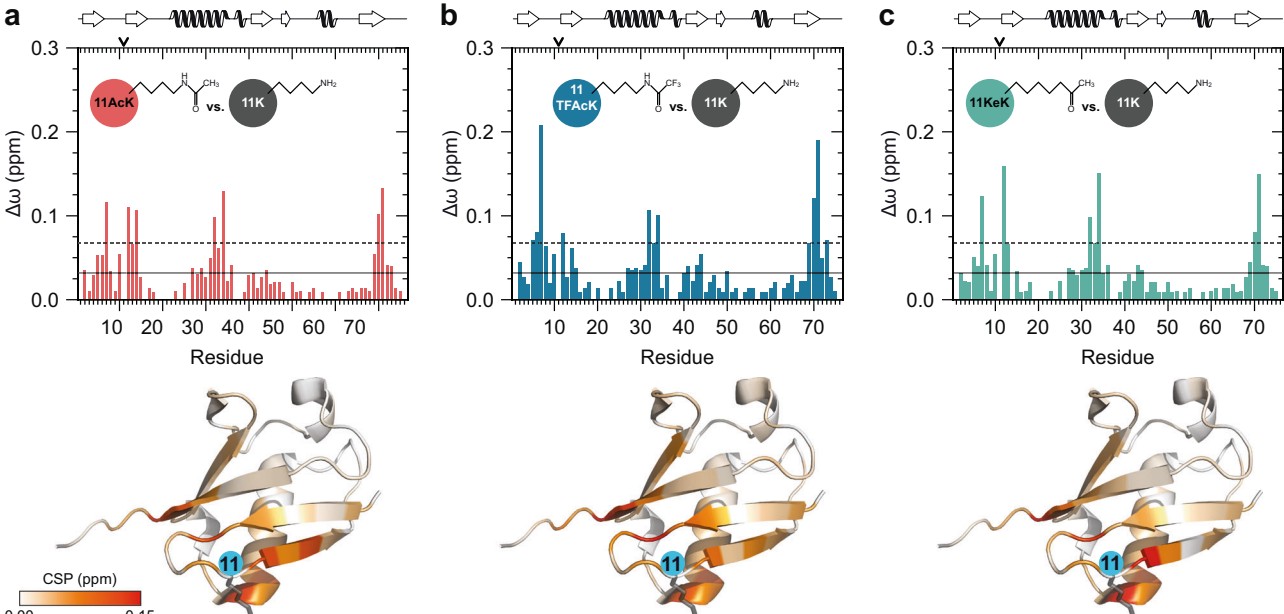

**Fig. 2 | Incorporation of AcK, TFAcK or KeK at position 11 of Ub results in similar structural changes. a–c** Weighted chemical shift perturbations (CSPs, Δω) of backbone amide resonances were calculated comparing the differently modified Ub variants (11AcK, 11TFAcK, 11KeK) with unmodified Ub (11K). The corresponding two-dimensional heteronuclear ¹H-¹⁵N HSQC NMR spectra are shown in Supplementary Fig. 2. In the upper panel, the corresponding CSP plots are presented for Ub 11AcK (**a**), Ub 11TFAcK (**b**), Ub 11KeK (**c**), while in the lower panel the respective CSP values are highlighted on the NMR solution structure of unmodified Ub (PDB ID 1D3Z)[63]. The Ub structure is illustrated in cartoon mode and the respective CSP values are indicated by using a continuous color palette ranging from white (Δω = 0.00 ppm) to red (Δω ≥ 0.15 ppm). Additionally, the position of K11 and its side chain are depicted by a blue circle and by anthracite sticks, respectively. In the CSP plots, the horizontal solid and dashed lines are indicating the mean and the mean plus one standard deviation, respectively, calculated over all CSP values from (**a–c**) excluding values related to position 11. On top of the plots, position 11 is indicated by an arrow and the secondary structural elements are schematically depicted according to the NMR solution structure of unmodified Ub. Source data are provided as a Source Data file.

The results obtained in HDM2 autoubiquitination assays support the notion that the effects of K11 acetylation are mediated by a combination of charge neutralization and structural changes induced by extension of the lysine side chain[11]. To corroborate this assumption and to show that incorporation of AcK, TFAcK or KeK induces similar structural changes, we performed high-resolution NMR spectroscopy using isotopically labeled variants of Ub 11AcK, 11TFAcK, and 11KeK. For all three Ub variants, two-dimensional ¹H-¹⁵N heteronuclear single quantum coherence (HSQC) NMR spectra were acquired (Supplementary Fig. 2), and weighted chemical shift perturbations (CSPs) were calculated for backbone amide resonances with reference to unmodified Ub (Fig. 2). By this, alterations in the chemical environment of corresponding amide proton and nitrogen spins as consequence of the conformational changes induced by the respective modification at position 11 of Ub can be monitored on a residue-by-residue basis. As expected by the length and composition of their side chains, Ub 11 AcK, 11TFAcK, and 11KeK show highly similar CSP patterns, including characteristic perturbations at the C-terminal end of the central α-helix of Ub[11]. In fact, only slight variations are apparent between the Ub variants in terms of amplitude and the exact position of the affected amino acids in the primary sequence.

Taken together, the above results indicate that both TFAcK and KeK properly mimic the effects of acetylation of K11 on Ub structure and that the actual side chain length at position 11 is the determining factor for the acetylation-induced structural changes resulting in the inefficient usage of the respective Ub variants by HDM2.

## Ub 11AcK, Ub 11TFAcK, and Ub 11KeK share similar interactomes
To further prove that TFAcK and KeK are functionally equivalent to AcK, we performed affinity enrichment coupled to high-resolution mass spectrometry (AE-MS) experiments with whole cell extracts to compare the protein-protein interaction properties (interactomes) of Ub 11AcK/TFAcK/KeK. As controls, we used Ub 11A, Ub 11R, Ub 11Q, and unmodified Ub (Ub 11K). In brief, we equipped all Ub variants with an N-terminal Strep tag II for immobilization on Strep-Tactin® beads. The immobilized Ub variants or Strep beads alone were incubated with whole cell extracts derived from HEK293T cells, and affinity-enriched proteins were identified by label-free quantitative MS. Significantly enriched proteins were determined by ANOVA statistics (S0 = 2, FDR = 0.005) (Fig. 3a).

Hierarchical clustering resulted in the identification of 338 significant interactors that bind to one or more Ub variants (Fig. 3b). Besides proteins that show no preference for any of the Ub variants (Cluster 4), we found 23 proteins that preferentially interact with Ub 11AcK, Ub 11TFAcK, and Ub 11KeK (Cluster 1) and 13 proteins that showed a strong decrease in binding to these Ub variants (Cluster 5). Similarly, proteins were identified that preferentially interact with Ub 11R and Ub 11K (Cluster 2, 3), suggesting that in this case the positive charge at position 11 is an important determinant for binding. To validate the AE-MS results, we analyzed some of the interactions by AE followed by western blot analysis (Supplementary Figs. 3a, 3b) and by using bacterially expressed proteins (NDP52[11], UCHL3[11]; Supplementary Fig. 3c). Reassuringly, we found a high correlation between the interaction patterns obtained by the different methods. For instance, NDP52 showed preferred binding to Ub 11TFAcK, Ub 11KeK, and Ub 11AcK, independent of the method used. Similarly, UCHL3 showed the strongest interaction with Ub 11K and 11R and the weakest interaction with Ub 11Q and Ub 11A.

The above results clearly indicate that the identity of the side chain at position 11 affects the protein interaction properties of Ub and that Ub variants with similar side chains have similar interaction properties (Fig. 3b). To further support this notion, we performed a Pearson correlation analysis, which indicates the degree of similarity or dissimilarity between two different data sets (Fig. 3c). Considering

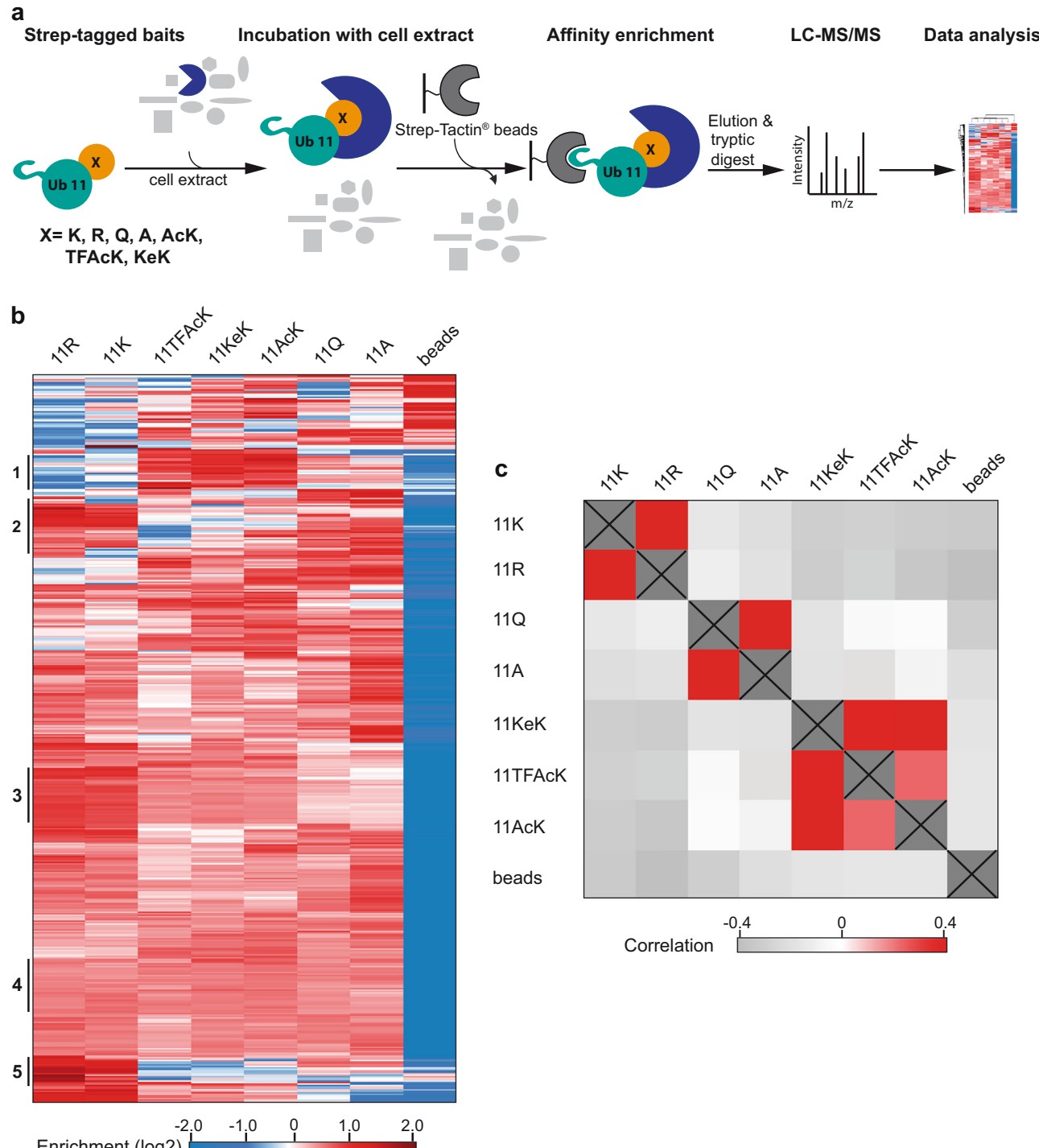

**Fig. 3 | Ub variants containing AcK or its analogs TFAcK and KeK resemble one another with respect to their protein interaction properties. a** Schematic overview of the AE-MS workflow for the identification of interactors of the Ub variants indicated. Unmodified Ub (11K) and each of the six Ub variants were used as bait molecules for affinity enrichment and incubated with HEK293T cell extract. Empty beads (beads) were used as control. **b** Hierarchical clustering of statistically significant interactors (rows) of the different bait molecules (columns). Red indicates enrichment, whereas blue indicates lack of enrichment. Thresholds for ANOVA statistics were set to S0 = 2 and FDR = 0.005. Cluster numbers, indicating binding to one or multiple Ub variants, are indicated on the left. **c** Correlation heatmap of significantly enriched interactors of the differently modified Ub variants. Strong correlation is indicated in red, no correlation in white, and inverse correlation in gray. Source data are provided as a Source Data file.

all 338 significant interactors, this revealed a high correlation between Ub 11R and Ub 11K and between Ub 11A and Ub 11Q, which reflects the significance of a positive charge at this position. Yet, there is no significant correlation of Ub 11A/11Q with Ub 11AcK, Ub 11TFAcK, and Ub 11KeK, while there is a high correlation between the latter three Ub variants in general and between Ub 11AcK und Ub 11KeK in particular (Fig. 3c; see also Supplementary Fig. 3d). This again indicates that the structural change induced by elongation of the side chain of K11 by acetylation has a major impact on the protein interaction properties of Ub and that this structural and, in

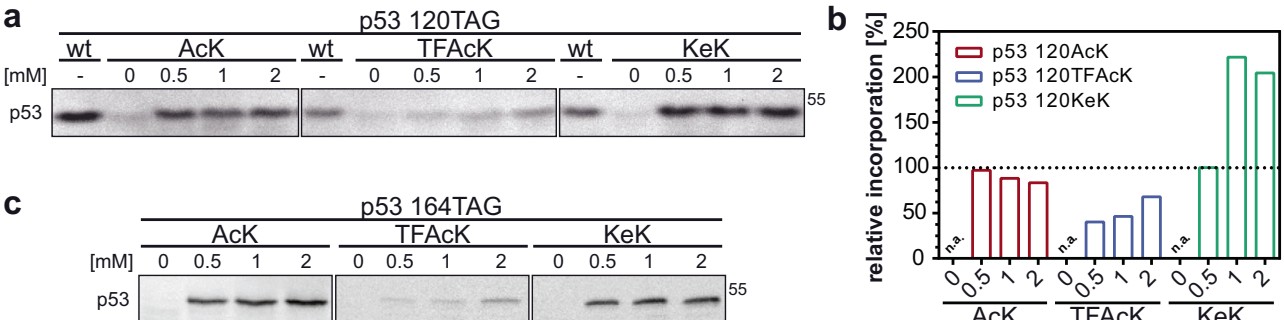

**Fig. 4 | Incorporation of AcK and its analogs TFAcK and KeK into p53 by genetic code expansion in H1299 cells. a** Incorporation of AcK, TFAcK, and KeK at position 120 of p53. **b** Quantification of relative p53 levels in the presence of AcK, TFAcK, and KeK shown in (**a**). Levels of wild-type p53 (wt) were set to 100 percent. **c** Incorporation of AcK, TFAcK, and KeK into position 164 of p53. H1299 cells were transfected with expression constructs encoding wild-type p53 (wt) or p53 with a TAG stop codon at position 120 and 164, respectively, and the AcK-RS/tRNA pair as described in Methods. Growth medium was supplemented with increasing concentrations of AcK, TFAcK, or KeK as indicated. 24 h upon transfection, p53 levels were determined by western blot analysis with the p53-specific DO-1 antibody (**a**, **c**). Running position of a molecular mass marker (55 kDa) is indicated. The experiments shown are representative of three independent experiments. Source data are provided as a Source Data file.

consequence, functional change is most reliably mimicked by TFAcK and KeK and not or only partially by Q.

Taken together, the results obtained show that Ub variants harboring TFAcK, KeK or AcK are highly similar in their propensity to interact with other proteins.

### Incorporation of AcK and its non-hydrolyzable analogs into p53 in human cells

From the results obtained with Ub, we reasoned that KeK and TFAcK are suitable AcK surrogates for studies in cells. Since Ub is a highly abundant protein in cells rendering analysis of ectopically expressed Ub rather difficult, we switched to the tumor suppressor p53 to test this assumption. p53 is subjected to various PTMs affecting its stability and/or its transcriptional activities resulting in different biological outcomes (e.g., cell cycle arrest, apoptosis, senescence) (summarized in refs. [24–27]). Most importantly for our purpose, the acetylation status of distinct K residues of p53, including K residues in its DNA binding domain (DBD), contributes to determining the selectivity or preference of p53 for certain target genes and thereby the eventual transcriptional program induced by p53[17–19].

To determine whether KeK and TFAcK are suitable AcK surrogates in cells, we generated expression constructs for human p53, in which the lysine codons at position 120 and/or 164 within the DBD were replaced by the amber stop codon TAG (p53-120TAG and p53-164TAG, respectively; note that K120 directly contacts DNA, while K164 does not). We chose these positions, since besides their physiological relevance[17–19] (see Discussion), incorporation of unnatural or non-canonical amino acids by ACS is rarely quantitative, resulting in a mixture of full-length and truncated versions of the POI. The more N-terminal the truncation occurs, the easier it is to distinguish between the full-length and the truncated protein, e.g., by western blot analysis. Furthermore, since in our case, the truncations occur in the DBD, the truncated p53 versions are not of functional relevance - they can neither bind to DNA nor oligomerize with full-length p53 - and, thus, do not need to be considered in the analysis.

The p53 expression constructs were transfected together with an expression construct encoding the CUA-tRNA as well as an aminoacyl-tRNA synthetase for AcK (AcK-RS)[28,29] and its analogs TFAcK and KeK (for further details, see "Methods") in the absence or presence of increasing concentrations of AcK, TFAcK or KeK. As cell line, we used H1299 cells, since they do not express endogenous p53[30]. As shown in Fig. 4, the three non-canonical amino acids were incorporated, resulting in the expression of full-length p53, with AcK and KeK being more efficiently used or incorporated into p53 than TFAcK (Fig. 4a–c).

Furthermore, the system is not limited to the incorporation of AcK or its analogs at a single position, but can also be used to generate p53 variants harboring two acetylated lysine residues (Supplementary Fig. 4), expanding the analytic potential of the system to multiple or at least double acetylated p53 variants.

### Incorporation of AcK, TFAcK or KeK has different effects on the transcriptional transactivation properties of p53

To test the functionality of the p53-120 variants, we determined their ability to transactivate the transcription of a reporter construct containing the binding site or response element (RE) for p53 of the *p21* gene[31,32]. As control, we used expression constructs encoding wild-type (wt) p53 and the tumor-derived mutant p53-273H (substitution of R273 by H), which has lost the sequence-specific DNA binding properties (ref. [33], and references therein). While the 120AcK and 120TFAcK variants were capable of inducing expression of the reporter construct, though with somewhat reduced efficiency as compared to wt p53, the 120KeK variant was not, or only poorly active (Fig. 5a). Moreover, the differences in the transactivational potential of the different p53 variants were also reflected in the levels of endogenous p21, indicating that the differences are not due to an artifact of the p21 reporter construct (Fig. 5b). Similar results were obtained with the p53-164 variants (Supplementary Fig. 5a, c).

At first glance, the finding that the KeK-containing p53 variants are significantly less active than the AcK and TFAcK variants seems to be surprising. However, it may be explained by the notion that in contrast to KeK, AcK and TFAcK are deacetylated in cells (ref. [15], and references therein) and, thus, the respective variants regain wt-like activity. To prove this possibility, we purified the different p53 variants from transfected cells and subjected them to Parallel Reaction Monitoring (PRM)-MS analysis[34]. The data obtained indeed suggest that the AcK and TFAcK variants are deacetylated, though to different extents, while the KeK variant is refractory to deacetylation as expected (Fig. 5c; and Supplementary Fig. 5d). The partial resistance of TFAcK is most likely due to the notion that Zn-dependent deacetylases can remove the fluorinated acetyl group[15,23], while NAD-dependent deacetylases cannot do so (see Discussion).

To exclude that the data obtained with KeK are the result of unwanted side effects, we performed several control experiments. Firstly, we looked into the possibility that the mere presence of KeK in the cell culture medium interferes with the transactivation activity of wt p53. Yet, a negative effect was not observed (Supplementary Fig. 5e). Secondly, p53-101KeK was similarly active as wt p53, p53-101AcK, and p53-101TFAcK, showing that incorporation of KeK per se

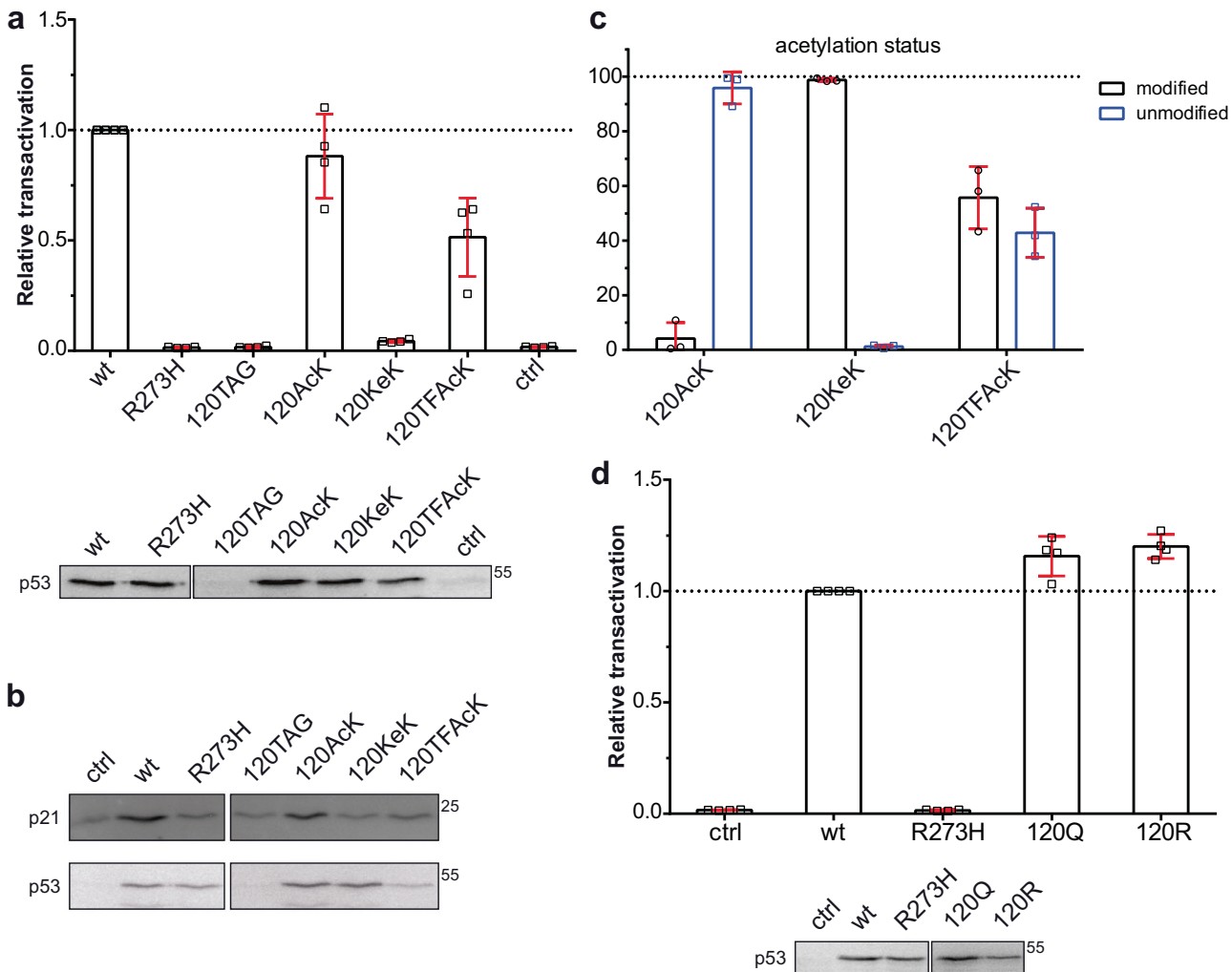

**Fig. 5 | Transactivation activity of p53 variants. a** H1299 cells were transfected with a p53-responsive expression construct encoding luciferase in the absence (ctrl) or presence of constructs encoding wild-type p53 (wt), the tumor-derived p53 mutant R273H, p53 with a TAG stop codon at position 120, and the AcK-RS/tRNA pair. In addition, cells transfected with the p53-120TAG construct were grown in the absence (120TAG) or presence of AcK (120AcK), TFAcK (120TFAcK) or KeK (120KeK). 24 h upon transfection, luciferase activity was determined as a measure for p53 activity and the relative values obtained were adjusted for transfection efficiency. Error bars indicate SD of the mean of four independent experiments. 20 percent of the extracts were subjected to western blot analysis using the p53-specific DO-1 antibody. **b** H1299 cells were transfected as in (**a**) in the absence of the p53-responsive reporter construct. 24 h upon transfection, levels of endogenous p21 protein and of ectopically expressed p53 were determined by western blot analysis using a p21-specific antibody and the p53-specific DO-1 antibody, respectively. **c** H1299 cells were transfected with expression constructs encoding p53-120TAG and the AcK-RS/tRNA pair. Cells were grown in the presence of AcK, KeK, and TFAcK as indicated and the presence of AcK, KeK, TFAcK or lysine (unmodified) at position 120 of p53 was determined by PRM analysis. Error bars indicate SD of the mean of three independent experiments. **d** Transactivation activity of the p53 variants 120Q and 120R was determined as described in (**a**). Error bars indicate SD of the mean of four independent experiments. Running position of molecular mass markers (25 kDa, 55 kDa) are indicated. Source data are provided as a Source Data file.

does not interfere with p53 function (Supplementary Fig 5b, c). Thirdly, we previously reported that in vitro, p53-120KeK binds to the p21 response element (p21-RE) with an efficiency similar to unmodified p53[35], indicating that incorporation of KeK does not disrupt the conformational integrity of the DBD of p53.

Finally, we asked whether or not Q can serve as substitute for AcK, when studying p53 function(s) in cells. This showed that unlike p53-120KeK, p53-120Q behaved like wt p53 in the p21 reporter assay (Fig. 5d), suggesting that in case of K120 acetylation, Q is not a suitable surrogate for AcK. In conclusion, our results obtained with Ub and p53 indicate that KeK is the most suitable AcK analog to study the effect of lysine acetylation of a POI in vitro and in cells.

## Discussion

Lysine acetylation is a prevalent PTM affecting many intracellular proteins[7,36–39]. Yet, determining the functional consequences as well as the physiological importance of acetylation for a POI remains a serious challenge. An obvious consequence of acetylation is neutralization of the positive charge of lysine's side chain, the functional significance of which can be determined by mutational analysis by replacing the respective lysine residue by arginine or an uncharged residue such as glutamine or alanine. An alternative, but not mutually exclusive possibility is that the eventual effect of acetylation is mediated by the extension of the length of lysine's side chain and/or the actual presence of the acetyl group. While this cannot be adequately addressed by conventional mutational analysis, the genetic code expansion technology in general and ACS in particular provide excellent tools to do so. Our comparative analysis of AcK and its analogs TFAcK and KeK and their effects on distinct properties of Ub and p53 strongly supports this proposition.

In support of the notion that the effects of acetylation are not necessarily explained by simple charge neutralization, we recently

reported that acetylation of Ub at K11 results in structural and, in consequence, functional changes that cannot be mimicked by substitution of K11 by Q[11]. Here, we extended the functional comparison of Ub 11AcK and Ub 11Q by determining their respective interactomes in vitro (Fig. 3). By comparison with unmodified Ub, Ub 11R, and Ub 11A, this clearly showed that even though the charge status at position 11 affects the Ub interactome (Fig. 3c), acetylation of K11 has additional or different effects on the interaction properties of Ub. Moreover, we show that incorporation of TFAcK and KeK at position 11 affects the structure of Ub in a manner similar, but not identical to AcK incorporation (Fig. 2). These slight differences, together with intrinsic variations in proteome-wide affinity enrichments, are reflected in highly similar, but not identical interactomes of Ub 11AcK, Ub 11TFAcK, and Ub 11KeK (Fig. 3, and Supplemental Fig. 3).

To determine the applicability of ACS-mediated incorporation of AcK and its analogs in eukaryotic cells, we switched to p53, since acetylation of distinct lysine residues in its DBD affects the transcriptional properties of p53[17–19] and since p53 activity can be readily monitored in cells by transcriptional reporter assays. Our results show that AcK, TFAcK, and KeK can be efficiently incorporated into p53 at different positions, with incorporation of TFAcK being somewhat less efficient than those of the others for unknown reasons (note that a similar observation was made for the efficiency of AcK and TFAcK incorporation by others[15]). However, only p53-120KeK was resistant to the action of deacetylases in cells, while p53-120AcK was quantitatively deacetylated and p53-120TFAcK was partially deacetylated (Fig. 5c). The partial resistance of TFAcK is likely due to the notion that Zn-dependent deacetylases can remove the fluorinated acetyl group, while NAD-dependent deacetylases cannot[15,23,40]. From a chemical perspective, TFAcK is more susceptible to nucleophilic attacks than AcK, as the trifluoroacetyl group is a better leaving group than the acetyl group[41]. However, while Zn-dependent deacetylases such as KDACs release the acetyl group via nucleophilic attack of the C atom of the carbonyl group by an activated water molecule, NAD-dependent deacetylases including sirtuins use NAD as electrophile for nucleophilic attack of the carbonyl oxygen atom[40]. Thus, due to the electron withdrawing effect of the three fluorine atoms, the nucleophilicity of the carbonyl oxygen is significantly decreased[41]. Along this line, it has recently been shown that HDAC2 is able to deacetylate FOXO4 variants harboring TFAcK or AcK at position 189, while Sirt2 can only deacetylate the AcK variant[15]. In contrast to TFAcK, KeK is resistant against the attack of Zn-dependent deacetylases as well[16,40]. An intermediate step in HDAC-catalyzed deacetylation is the nucleophilic attack of the carbonyl C atom by a water molecule. To do so, a base, presumably a histidine residue, accepts a proton from the water molecule and transfers it to the nitrogen atom of the amide bond, which facilitates the final cleavage of the amide bond[40]. In contrast to the nitrogen, the carbon atom of KeK (Fig. 1a) can most probably not function as an acceptor for the proton. Thus, while in the case of KeK the nucleophilic attack may still occur, though with reduced efficiency, cleavage of the C-C bond is highly unlikely. In any case, this difference in the deacetylation susceptibility of AcK, TFAcK, and KeK may represent an attractive possibility to dissect the contribution of the different deacetylase families to the deacetylation of a POI.

The difference in acetylation status of the different p53-120 variants is also reflected in different abilities to transactivate a p21-reporter construct as well as to induce expression of endogenous p21 (Fig. 5). p53-120AcK is similarly active as wt p53, p53-120TFAcK is somewhat less active, and p53-120KeK does neither detectably transactivate the reporter construct nor induce endogenous p21 expression. In addition and in line with published data[42], p53-120R (substitution of K by R) efficiently transactivated the reporter construct. Moreover, the 120Q variant behaved like wt p53 in the reporter assay, suggesting that in this case, charge neutralization does not suffice to mimic the effect of acetylation. Our data with p53-120KeK are

in line with ChIP data indicating that within cells, p53 acetylated at K120 does not accumulate at the *p21* gene[43] and with the notion that acetylation of K120 does not result in the induction of expression of the *p21* gene (refs. 18,19, and references therein). Yet, similar to our data with p53-120KeK[35], it has previously been shown that in vitro, p53-120AcK generated by the ACS method binds to the p21-RE with an affinity similar to unmodified p53[32]. How can these apparent contradictory data be reconciled? In a simplified view, cell cycle arrest target genes such as *p21* contain high affinity REs for p53, while pro-apoptotic target genes contain low affinity REs (ref. 42, and references therein). Furthermore, there is data to indicate that p53 contacts high affinity REs and low affinity REs in a distinct manner and that binding of p53-120AcK does not affect the conformation of high affinity REs, but changes the conformation of low affinity REs[32,42,44]. Moreover, there is evidence that the conformation of p53-120AcK bound to high affinity REs is different to the one bound to low affinity REs[32]. Such conformational changes that depend on both p53 acetylation and binding of p53 to DNA in a sequence-specific manner may affect the protein-protein interaction properties of p53 and thereby its ability to differentially transactivate p53 target genes.

Whereas K120 acetylation has been associated with the apoptotic program of p53 by inducing expression of pro-apoptotic genes such as *BAX* and *PUMA*, K164 acetylation has been involved in the cell cycle program by inducing the expression of cell cycle arrest genes including *p21*[17–19,43,45]. Yet, p53-120KeK or p53-164KeK were inactive in our reporter assays, irrespective of whether a p21 reporter construct or a PUMA reporter construct was used (Fig. 5, and Supplementary Figs. 5 and 6). As discussed above, all of our data indicate that KeK structurally and functionally resembles AcK and, thus, its incorporation should affect p53 activity in a manner similar to lysine acetylation. Yet, we cannot exclude that proteins exist that bind to KeK with lower affinity than to AcK. For instance, bromodomains are known binders of AcK-containing proteins[46]. Thus, there may be bromodomain-containing proteins or other proteins that specifically bind to p53-120AcK, and such interactions may be critically involved in p53-mediated transactivation of the p21 and/or PUMA reporter constructs. However, this does not appear to be the case, since p53-120R efficiently induced expression of both the p21 reporter construct and the PUMA reporter construct. Irrespectively, a likely conclusion of the data obtained with p53-120KeK (p53-TFAcK), p53-120Q, and p53-120R is that a positive charge at position 120 is not critical for p53-mediated transcription of the p21 reporter construct and, thus, the negative effect of K120 acetylation cannot be explained by charge neutralization. For the PUMA reporter construct, charge neutralization at position 120 contributes to the negative effect of KeK incorporation, but additional properties of KeK (and thus, AcK) appear to be involved. Taken together, our results obtained with the different p53 variants indicate that a comparative analysis of POI variants, in which a distinct K residue is replaced by R, Q, or AcK surrogates, provides critical information about if and how acetylation affects known properties/functions of a POI. Because it is not hydrolyzable, KeK represents a well-suited AcK mimic for such studies in cells, with the potential limitation that in some cases, protein(s) critically involved in executing the functional consequences of acetylation may recognize KeK less efficiently than AcK. For in vitro studies, we propose to use AcK as long as hydrolysis is not a matter of concern[11,47].

A possibility to explain the apparent discrepancy between our data and published data is that our reporter constructs miss additional sequence elements of the respective endogenous genes. At least in case of p21, this seems unlikely, since endogenous p21 expression was also not induced by p53-164KeK (Supplementary Fig. 5c). Furthermore, many of the published data have been obtained by mutational analysis, for instance by replacing respective lysine residues by arginine (summarized in refs. 16–18). Thus, some of the effects ascribed to lysine acetylation may in part be due

to other lysine modifications such as mono-ubiquitination[48,49] or lactylation[50]. Other possibilities would be that mono-acetylation of p53 or acetylation in general is not sufficient for the induction of certain target genes. In other words, it seems likely that a combination of acetylation with additional PTMs such as phosphorylation is required to observe the respective transcriptional effects, a scenario that may not be achieved in our experimental set-up. For instance, it could be envisioned that such phosphorylation events occur only under certain stress conditions. An alternative, but not mutual exclusive possibility is that to be functional, such PTMs have to occur in a timely ordered, i.e., consecutive manner. Along this line, prior lysine acetylation, in particular incorporation of an AcK mimic that cannot be hydrolyzed as in our case, may prevent phosphorylation of p53 at residues required for transcriptional transactivation. To address such possibilities, the continuous development and refinement of the genetic code expansion technology to allow the combinatorial incorporation of different non-canonical amino acids resembling non-reversible, yet faithful analogs of respective PTMs should prove highly valuable.

# Methods

### Synthesis of 2-amino-8-oxononanoic acid (Ketolysine; KeK)

KeK was synthesized as described previously[16], except for the final purification step. Instead of using a Dowex column, KeK was purified twice via MPLC and then lyophilized. For MPLC purification, a PrepChrom C-700 (Büchi) with SuperVarioFlash® D26 RP18 30 g or D40 RP18 90 g (Götec-Labortechnik GmbH) were used. After sample application, the column was washed with $H_2O$ for 20 min and then a linear gradient (0%-100% acetonitrile in 20 min) of acetonitrile in $H_2O$ was applied with a flow rate of 20 ml/min. Runs were monitored by UV detection at 254 nm. Fractions containing KeK were pooled and lyophilized.

To record NMR spectra of KeK, a Bruker Avance III 400 (400 MHz) was used. Spectra were measured in the presence of $D_2O$ (the deuterium resonance signal is used as internal reference frequency for other nuclei) at room temperature. The program MestReNova 14.2 was used for spectrum evaluation. Shifts are indicated in ppm (σ-scale) and coupling constants in Hertz (Hz). Source data are provided as a Source Data file.

$^1H$ NMR (400 MHz, $D_2O$): δ [ppm] = 3.88 (t, J = 6.1 Hz, 1H, H-1), 2.62 (t, J = 7.3 Hz, 2H, H-6) 2.25 (s, 3H, CH3), 2.06-1.82 (m, 2H, H-2), 1.61 (p, J = 7.4 Hz, 2H, H-5), 1.53-1.31 (m, 4H, H-3 & H-4).

$^{13}C$ NMR (101 MHz, $D_2O$): δ [ppm] = 217.56 (CO), 174.68 (CO2H), 54.65 (C-1), 43.13 (C-6), 30.20 (C-2), 29.31 (CH3), 27.88 (C-4), 24.08 (C-3), 22.98 (C-5).

### Plasmids

Codon-optimized cDNA encoding Ub with a C-terminal His6-tag was inserted into the pGEX-2TK backbone replacing the GST cDNA to yield pKS Ub-His[51]. cDNAs of Ub variants, in which the amber stop codon (TAG), a glutamine, an alanine or arginine codon replaced the respective lysine codon, were generated via site-directed mutagenesis[11].

The cDNA encoding the aminoacyl-tRNA synthetase for AcK (AcK-RS) derived from *Methanomethylophilus alvus* was kindly provided by Dr. K. Lang (ETH Zürich) and was inserted into the pRSFDuet vector (Merck Millipore) for bacterial expression[11]. Plasmids encoding tRNA and AcK-RS derived from *Methanosarcina mazei* for incorporation of AcK/TFAcK/KeK in H1299 cells were kindly provided by Dr. Simon Elsässer (Karolinska Institute, Stockholm)[28,29].

cDNAs encoding wild-type p53 or the tumor-derived mutant 273H with and without C-terminal Strep tag II was inserted into pcDNA3. cDNAs for p53 variants, in which the TAG stop codon, a glutamine or an arginine codon replaced the respective lysine codon, were generated via site-directed mutagenesis.

Reporter constructs encoding luciferase were generated by introducing p53-responsive elements (p21, PUMA)[32] into the pGL4 luciferase reporter system.

### Bacterial expression and protein purification

Human His6-tagged UBA1[51,52], N-terminal His6-tagged UCHL3[51], and C-terminal His6-tagged UbcH5b[51,53] were expressed and purified as described.

Expression of the various GST fusion proteins was performed in *E. coli* BL21 (DE3) Rosetta. GST-SUMO fusion proteins of HA-tagged HDM2 and NDP52 were expressed and purified as described[11].

### Bacterial generation of modified Ub and unmodified Ub

For generation of Ub site-specifically modified at position 11 (with and without an N-terminal Strep tag II), the respective pKS vector was co-transformed with pRSFDuet harboring the AcK-RS/t-RNA pair for amber codon suppression into *E. coli* B834 (DE3) cells. Cells were grown at 37 °C in 100 ml LB medium containing 100 μg/l carbenicillin/ 34 μg/l kanamycin until $OD_{600} = 1$ was reached. Then, cultures were diluted to an $OD_{600} = 0.1$ with minimal media devoid of phenylalanine and further incubated. At an $OD_{600} = 0.4$, either 10 mM AcK (abcr) or 20 mM TFAcK (abcr) or 10 mM KeK (racemate) and 20 mM nicotinamide (Sigma-Aldrich) were added. Gene expression was induced at $OD_{600} = 0.7$ by the addition of 1 mM IPTG (final conc.). After incubation at 25 °C for 16 h, cells were harvested by centrifugation, resuspended in lysis buffer (50 mM Tris-HCl pH 7.4, 150 mM NaCl, 30 mM imidazole, 1 % Triton X-100, 1 mg/ml aprotinin/leupeptin, 1 mg/ml Pefabloc) and lysed by sonication. The lysate was cleared, and modified Ub-His was purified using a 5 ml HisTrap HP (Äkta pure25, GE Healthcare) with a step from 30 mM to 242 mM imidazole in 50 mM Tris-HCl pH 7.4, 150 mM NaCl for elution. Protein-containing fractions were pooled, UCHL3 was added to a final concentration of 75 μg/ml to remove the C-terminal His-tag, and the samples were incubated for 4 h. After cleavage, modified Ub variants were dialyzed against 25 mM acetate pH 4.5 overnight at 4 °C. Ub samples were further purified via a 1 ml HiTrap SP HP (GE Healthcare) with a linear gradient from 25 mM acetate pH 4.5 to 25 mM acetate 4.5, 1 M NaCl. Modified Ub containing fractions were pooled and dialyzed against 25 mM Tris-HCl pH 7.5, 50 mM NaCl. The concentration was determined with the BCA Protein Assay Kit (Thermo) and commercial Ub (Sigma-Aldrich) as standard. The correct masses of the modified Ub variants were confirmed by ESI-MS (Supplementary Fig. 1).

Expression and purification of unmodified Ub and the Ub variants containing Q, A or R instead of K at position 11 were performed similar to the modified Ub variants, except that AcK/TFAcK/KeK and nicotinamide were not added to the respective expression cultures.

### In vitro autoubiquitination assay

For in vitro autoubiquitination, 100 nM UBA1, 0.5 μM UbcH5b, and 0.55 μM HDM2 were incubated in 25 mM Tris-HCl pH 7.5, 50 mM NaCl, 1 mM DTT, 2 mM ATP, and 2 mM $MgCl_2$. The reaction was started by addition of 7.8 μM (final conc.) of the respective Ub variant and incubated at 37 °C for the times indicated. Reactions were stopped by the addition of 5x reducing SDS loading buffer. Entire reaction mixtures were electrophoresed in 12.5% SDS-polyacrylamide (SDS-PA) gels followed by Coomassie blue staining. Quantification was performed with Fiji ImageJ (Version 1.51).

### Generation of isotopically labeled Ub for NMR analysis

For preparation of $^{15}N$-labeled proteins, *E. coli* harboring the expression construct for Ub with a TAG stop codon at position 11 was grown in LB medium at 37 °C until $OD_{600} = 1$. This culture was used to inoculate $^{15}N$ M9 medium (33.7 mM $Na_2HPO_4$, 22 mM $KH_2PO_4$, 8.55 mM NaCl, 9.35 mM $^{15}NH_4Cl$, 1 mM $MgSO_4$, 0.3 mM $CaCl_2$, 134 μM EDTA, 31 μM $FeCl_3$, 6.2 mM $ZnCl_2$, 0.76 μM $CuCl_2$, 0.42 μM $CoCl_2$, 1.62 μM

$H_3BO_3$, 0.081 µM $MnCl_2$, 1 mg/l biotin, 1 mg/l thiamine, 0.4% glucose (w/v)). Upon growth overnight at 37 °C, 100 ml of the culture were diluted with 900 ml of pre-warmed $^{15}N$ M9 medium. Cells were grown at 37 °C until $OD_{600} = 0.4$ was reached, and then either 10 mM AcK (abcr), 20 mM TFAcK (abcr) or 10 mM KeK (racemate) and 20 mM nicotinamide (Sigma-Aldrich) (final conc.) were added. Gene expression was induced at $OD_{600} = 0.7$ by the addition of 1 mM IPTG (final conc.). After incubation at 25 °C for 16 h, cells were harvested by centrifugation. Purification of isotopically labeled Ub was performed as described above for unlabeled Ub.

## Acquisition of NMR spectroscopic data
NMR experiments were performed at $T = 298$ K on a Bruker Avance Neo 800 MHz spectrometer equipped with a quadruple (QCI) cryo probe with z-gradient. Datasets were processed using NMRPipe[54] and analyzed by NMRView[55]. Samples were prepared in 20 mM $NaH_2PO_4$ (pH 6.8) supplemented with 5% (v/v) $D_2O$, which serves as frequency lock. All measurements were performed with singly $^{15}N$-labeled variants of Ub 11AcK, 11TFAcK and 11KeK using protein concentrations of 33 µM, 500 µM and 60 µM, respectively. For each protein, a two-dimensional heteronuclear $^1H$-$^{15}N$ HSQC spectrum was collected with 1024 and 256 increments in the direct $^1H$ and indirect $^{15}N$ dimension, respectively, using 8-32 transients depending on the protein concentration. WATERGATE pulse sequence and presaturation were applied for efficient solvent suppression.

## Backbone resonance assignment and calculation of chemical shift perturbations
The backbone resonance assignment of Ub 11AcK was transferred from ref. 11 and served as basis for the assignment of backbone amide resonances of Ub 11TFAcK and 11KeK. Due to the slight differences in cross peak position, this could be successfully accomplished by visual estimate when the two-dimensional $^1H$-$^{15}N$ HSQC NMR spectra of Ub 11TFAcK and 11KeK were directly compared to Ub 11AcK.

Weighted chemical shift perturbations between the different Ub variants and unmodified Ub were calculated according to the following equation[56]:

$$\Delta\omega = \sqrt{\frac{\left(\Delta^1H\right)^2 + \frac{1}{25}\left(\Delta^{15}N\right)^2}{2}} \quad (1)$$

where $\Delta^1H$ is the cross peak shift in proton and $\Delta^{15}N$ is the cross peak shift in nitrogen dimension, respectively, between two corresponding cross peaks. Note that the assignment of backbone resonances arising in the two-dimensional heteronuclear $^1H$-$^{15}N$ HSQC NMR spectrum related to unmodified Ub was also taken from ref. 11.

## Cell culture and extract preparation
HEK293T and H1299 cells were cultured in Dulbecco's Modified Eagle's Medium (DMEM, Thermo Fisher) supplemented with 10% fetal bovine serum (FBS, Sigma) at 37 °C and 5% $CO_2$ atmosphere.

For affinity enrichment experiments, HEK293T and H1299 cell pellets were resuspended in cell lysis buffer (1x PBS, 2 mM $MgCl_2$, 1 mM DTT, supplemented with 1 mg/ml aprotinin/leupeptin, 1 mg/ml Pefabloc) and subsequently lysed by sonication. After centrifugation (21,000x g, 30 min, 4 °C), the total protein concentration of the supernatant was determined with the BCA Protein Assay Kit (Thermo Fisher) and freshly dissolved BSA as standard.

For transactivation studies, H1299 cell pellets were resuspended in lysis buffer (100 mM Tris-HCl pH 8.0, 100 mM NaCl, 1 mM DTT, 1% (v/v) Nonidet P-40, supplemented with 1 mg/ml aprotinin/leupeptin, 1 mg/ml Pefabloc) and incubated for 30 min at 4 °C prior to centrifugation (16,100x g, 30 min, 4 °C). Cleared whole cell extracts were analyzed by western blot and luciferase reporter assay as described below.

## Affinity enrichment
For affinity enrichment of Ub interacting proteins, 25 µg of each Strep-Ub variant was incubated with 2.5 mg whole cell lysate for 10 min on ice. After washing 50 µl Strep-Tactin® beads (iba) with three times 200 µl cell lysis buffer, the mixture was added to the beads. Incubation at room temperature for 3 h in an overhead shaker was followed by 5 washing steps with 200 µl cell lysis buffer each. Bound proteins were eluted by incubating the beads for 5 min on ice with two times 150 µl and one time 100 µl elution buffer (100 mM Tris-HCl pH 8.0, 150 mM NaCl, 1 mM EDTA, 2.5 mM desthiobiotin). The eluates were combined and freeze-dried. Each experiment was performed in triplicates.

The freeze-dried eluate was resuspended in 100 µl 8 M urea, reduced with 5 mM TCEP for 30 min at 37 °C and alkylated with 10 mM iodoacetamide for 30 min at room temperature. After dilution to 1 M urea with 50 mM $NH_4HCO_3$, 5 µg trypsin were added, and proteins were digested for 20 h at 37 °C. The digested samples were freeze-dried and kept at -20 °C until they were prepared for MS measurement.

Prior to MS measurement, the tryptic peptides were dissolved in 50 µl 0.1% TFA in $H_2O$, acidified with 10% TFA and desalted using Pierce™ C18 spin tips (Thermo Fisher).

For affinity enrichment of ectopically expressed Strep-tagged p53, transiently transfected cells were harvested and lysed as described above. Whole cell extracts were added to 20 µL Strep-Tactin® XT beads (iba) pre-equilibrated in lysis buffer. After incubation at 4 °C for 3 h in an overhead shaker, beads were washed 5 times with 200 µl lysis buffer each. Bound proteins were eluted as stated above for Ub with elution buffer (100 mM Tris-HCl, pH 8.0, 150 mM NaCl, 1 mM EDTA, 50 mM biotin) and lyophilized. Freeze dried eluates were resuspended in SDS loading buffer and subjected to SDS-PA gel electrophoresis for in-gel digestion. SDS-PA gels were stained by colloidal Coomassie blue, and gel pieces containing p53 were excised and digested according to ref. 57. In brief, gel pieces were washed in 100 mM $NH_4HCO_3$ for 10 min, destained in 50 mM $NH_4HCO_3$, 50% (v/v) acetonitrile and subsequently dehydrated in pure acetonitrile. Disulfide bonds were reduced by addition of 10 mM TCEP in 100 mM $NH_4HCO_3$ for 60 min at 56 °C followed by dehydration in pure acetonitrile and subsequent alkylation in 50 mM iodoacetamide in 100 mM $NH_4HCO_3$ for 60 min at room temperature in the dark. Following dehydration in acetonitrile, 5 µg trypsin were added, and proteins digested for 20 h at 37 °C. Peptides were extracted in 30% acetonitrile, 5% formic acid and 60% acetonitrile, 5% formic acid. Combined extracts were freeze dried and kept at -20 °C until preparation for MS measurements. Prior to MS measurement, the tryptic peptides were dissolved and desalted as stated above.

## Mass spectrometric analysis
Peptide samples were analyzed as previously described[11] using a Q-Exactive HF mass spectrometer (Thermo Fisher Scientific, Bremen, Germany) coupled to an Easy-nLC 1200 ultrahigh-pressure nanoflow chromatography platform (Thermo Scientific, Odense, Denmark). Samples were solubilized in 0.1% formic acid and transferred onto a C18 reversed-phase analytical column (75 µm × 15 cm). Peptide species were separated at 300 nl/min employing a linear elution profile ranging from 6–40% solvent B (0.1% formic acid in 80% acetonitrile) over 165 min. Data-dependent MS acquisition was performed with MS1 scans spanning 350–1500 m/z in the Orbitrap analyzer at a resolving power of 60,000 at 200 m/z, using an AGC target of 3e6 and a maximum ion accumulation time of 60 ms. The 10 most abundant precursor ions were selected for subsequent MS/MS interrogation. Only precursor ions with charge states between 2 and 6 were considered, and dynamic exclusion was maintained at 30 sec. Fragmentation was achieved via higher-energy collisional dissociation (HCD) with a normalized collision energy of 28%. Fragment ion spectra were collected at a resolution of 15,000 with an AGC target of 1e5 and a

maximum injection time of 60 ms. Each biological replicate was analyzed in two separate technical runs.

For label-free quantification, raw LC-MS/MS datasets were processed in MaxQuant[58] (version 1.6.8) using standard parameters, enabling both match-between-runs and LFQ quantification. Protein identification was performed against the human reference proteome obtained from UniProt (download date: 2018-02-22). Subsequent data processing was performed with Perseus[59] (version 1.6.10.50). Proteins annotated as reverse hits or common contaminants were removed. LFQ intensity values were transformed using log2 scaling. Only proteins detected in at least 4 of the 6 measurements (three biological replicates, each analyzed in technical duplicate) were retained for downstream evaluation. Missing values were imputed from a normal distribution (width = 0.3, shift = 1.2) under the assumption that these proteins were present but below the detection limit. Proteins exhibiting significant enrichment were determined using ANOVA (S0 = 2, FDR = 0.005) and subsequently Z-score normalized. After computing group medians across replicates, enriched proteins were clustered based on correlation patterns and visualized as heatmaps. Additional annotations (GO annotations, KEGG, Pfam) and identification of enriched terms was done with Perseus, as described previously[60]. The top 100 significantly enriched proteins were additionally used to compare interactors between Ub 11AcK and Ub 11KeK and were plotted as Venn diagram.

Intact protein mass measurements were performed on a micrO-TOF II (Bruker) interfaced with an Agilent 1260 Infinity II liquid chromatography system. Proteins were resolved on an analytical column (Nucleodur 300-5 C4 ec, EC 150/4, Macherey-Nagel) at a flow rate of 300 μl/min using a linear gradient from 5% to 100% solvent B (0.1% formic acid in acetonitrile) over 15 min. The electrospray ion source was operated in positive ionization mode, scan range 650 – 2100 m/z at 1 Hz acquisition rate and 2x rolling average was used. Re-calibration and spectra deconvolution was performed by using Compass Data Analysis software (Bruker).

To quantify the p53 modification status, parallel reaction monitoring (PRM) using a precursor inclusion list and targeted MS2 mode was employed. Peptides were separated on a 15 cm Acclaim PepMap C18 column (P/N 164943, Thermo Scientific) across a 50 min gradient from 2% to 45% solvent B (0.1% formic acid in 80% acetonitrile). A full mass spectrum was acquired on an Orbitrap Tribrid Fusion (Thermo Fisher Scientific, Bremen, Germany) at 120 K resolution over a mass range of 330–1250 m/z (adapted to precursors in inclusion list). The maximum injection time was set to 50 ms and an AGC target of 4e5. Precursors were isolated sequentially according to the inclusion list with an isolation window width of 1.6 m/z and fragmented by HCD at 27% NCE. Fragment mass spectra were acquired in the Orbitrap at 30 K resolution and an AGC target of 5e4. PRM data were analyzed via Skyline software 23.1.0.268. MS1-filtering isotope precursor ion peak was set to 3 and peptides were quantified on the MS2-level. Obtained values were plotted with GraphPad Prism 6.01.

## In vitro binding assay with recombinant proteins

Verifications of the protein interactions with the respective Strep-Ub variants were performed as described previously[11]. In brief, 5 μM of the respective Strep-Ub variant were incubated 10 min on ice in binding buffer (1x PBS, 10 mM MgCl2, 5 mM DTT) with 5 μM of recombinant UCHL3 or NDP52. Then, 5 μl Strep-Tactin® beads (iba) pre-equilibrated in binding buffer were added and the mixture incubated for 3 h at room temperature. After washing, bound proteins were eluted twice by incubating the beads with 30 μl elution buffer (200 mM Tris-HCl pH 8.0, 300 mM NaCl, 4 mM EDTA, 10 mM desthiobiotin). Eluates were combined, subjected to SDS-PA gel electrophoresis, and analyzed by Coomassie blue staining and/or western blot analysis.

## Transient transfection and incorporation of AcK, TFAcK, and KeK into p53

$2 – 3 \times 10^5$ cells were seeded in 6-well plates (Sarstedt) 24 h before transfection. Cells were transfected at an approximate confluency of 80% with Lipofectamine 2000 (Invitrogen-Thermo Fisher Scientific) according to manufacturer's instructions with plasmids encoding the AcK-RS/tRNA pair[28,29] required for incorporation of AcK and its analogs TFAcK and KeK and an expression construct for p53 harboring a TAG stop codon at the position indicated. In addition, an expression construct encoding β-galactosidase was cotransfected to determine relative transfection efficiencies. Unless stated otherwise, 1 mM AcK, 8 mM TFAcK or 1 mM KeK (final conc.) were added at the time of transfection, and cells were harvested after 24 hours. After cell lysis, cleared whole cell extracts were subjected to β-galactosidase assay and β-galactosidase normalized aliquots were electrophoresed on 12.5% SDS-PA gels followed by western blot analysis with an anti-p53 antibody (DO-I) and an HRP-coupled anti-mouse secondary antibody. Uncropped and unprocessed blots are provided as a Source Data file. The presence of the AcK-RS/tRNA pair and of AcK, TFAcK or KeK did not affect the viability of transfected cells as determined by crystal violet staining[61].

## Luciferase reporter assays

Cells were transfected as stated above with expression constructs for the p53 variants indicated, the AcK-RS/tRNA pair (if required), and β-galactosidase, and a p53-responsive luciferase reporter construct (p21, PUMA). 24 h upon transfection, whole cell extracts were prepared and 10 μl aliquots of each extract were transferred in duplicates to a black 96-well plate and luciferase activity automatically determined by a micro-plate reader (Wallac 1420 VICTOR³ Multilabel plate reader, PerkinElmer). In brief, 100 μL of luciferase reaction mix (470 μM D-luciferin, 530 μM ATP, 270 μM CoA, 200 μM EDTA, 20 mM tricine, 2.67 mM MgSO4, 1.07 mM MgCO3, 33.3 mM DTT, pH 7.5) were dispensed into each well and the reaction mixture incubated for 5 min. Luciferase activity was assessed by measuring light emission for 10 seconds.

## Reporting summary

Further information on research design is available in the Nature Portfolio Reporting Summary linked to this article.

## Data availability

The NMR solution structure and the crystal structure of ubiquitin used in this study are available in the ProteinDataBank under the accession codes 1D3Z and 1UBQ, respectively. The mass spectrometry proteomics data have been deposited to the ProteomeXchange-Consortium via the PRIDE repository[62] identifiers PXD056021 for the PRM data (http://proteomecentral.proteomexchange.org/cgi/GetDataset?ID = PXD056021) and PXD056063 for the AE-MS data (http://proteomecentral.proteomexchange.org/cgi/GetDataset?ID = PXD056063). Other data generated in this study are provided in the Supplementary Information and in Source Data files. Source data are provided with this paper.

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

## Acknowledgements

We thank the Proteomics Center of the University of Konstanz for assistance with mass spectrometric experiments, and Silke Büstorf and Nicole Richter-Müller for technical support. We are grateful to Dr. Kathrin Lang and Dr. Simon Elsässer for providing cDNAs and plasmids encoding the acetyllysyl-tRNA synthetase and the corresponding tRNA. This work was supported by the DFG (SFB969, Projects B3, A.M. and M.Sch., and B9, M.K.; TRR353/1 - 471011418, F.S.). S.M.K. thanks the doctoral fund of the University of Konstanz and the Konstanz Research School Chemical Biology for financial support. K.S. acknowledges the Zukunftskolleg of the University of Konstanz for a doctoral fellowship. F.S. is grateful for funding by the DFG (project grants 516836828 and 496470458).

## Author contributions

S.M.K., M.S., and M.Sch. conceived the study and experimental approach, S.M.K. generated all Ub variants, K.S. provided E1, E2, and UCHL3, and established the AE-MS workflow for Ub variants, M.S. generated all p53 expression constructs, J.L. synthesized KeK. S.M.K. performed HDM2 autoubiquitination assays, AE-MS with the Ub variants, and interaction studies for AE-MS validations. T.S. conducted NMR spectroscopic analyses. M.S. performed all transient transfection experiments with p53. J.J. performed the PRM analysis. S.M.K., M.S., T.S., J.L., J.J., F.S., A.M., M.K. and M.S. analyzed the data. S.M.K., M.S. and M.Sch. wrote the paper. All authors provided critical feedback on the paper.

## Funding

## Competing interests

The authors declare no competing interests.
