## [Transparent Peer Review file · Nature Communications]

Non-hydrolyzable acetyllysine analogs to study protein acetylation in vitro and in cells

Corresponding Author: Professor Martin Scheffner

Version 0:

Reviewer comments:

Reviewer #1

(Remarks to the Author)

The manuscript: Non-hydrolyzable acetyllysine analogs to study protein acetylation in vitro and in cells presents a very nice study of a lysine substitute that mimics acetylation using a unnatural amino acids, one of which is chemically resistant to deacetylation. This is a significant improvement in the existing strategy to study acetylated lysine and I expect that this chemically resistant derivative will become a significant and important tool in molecular biology with many broad applications.

Two protein host systems are used to test the synthetic acetylated lysine derivatives: Ubiquitin and p53 – both extremely relevant proteins in cell biology. They use the now standard strategy of stop codon inserted into the gene with complementary tRNAs charged with the unnatural lysine to incorporate these lysine derivatives. Two new derivatives include a lysine with a fluorinated acetyl methyl group TFACK and another missing the terminal side chain nitrogen that normally forms an amide bond with the acetyl group, KeK. The side chain nitrogen makes the attachment of the acetyl group chemically labile; removing this group was a very smart idea. The experimental strategies employed here are designed to reveal significant information. For example, NMR chemical shift mapping is exquisitely sensitive to small changes in structure that might be induced by amino acid changes. The authors asked the question of whether the lysine derivatives affected protein-protein interactions of Ubiquitin in (Figure 3). This is an excellent assay for functional consequences in comparing acetylation mimics.

Their study of p53 reveal the usefulness of the KeK derivative as well as limitations of their TFACK version. Problems with TFACK include poor incorporation into p53 and a higher propensity for deacetylation compared to KeK (which appears quite stable).

Thus, we have a winner! KeK appears to be a superior substitution to study acetylated lysine.

In addition to demonstrating the usefulness of the KeK variant, their results show clearly that glutamine is an insufficient mimic of a neutralized (acetylated) lysine. This was proven by independent assays in both the model systems of study (Ubiquitin and p53). These data should serve as cautionary evidence for those researchers exploring the function of modified lysines: be very careful when substituting in glutamine.

The paper is well-written, the methods are detailed, and the strategy is sound.

(Very) minor suggestions for correction:

1. The terminology for amino acids varies between single letter codes and spelling out the amino acid name. It would be helpful to use consistent terms.
2. Organization: The last paragraph of the Introduction contains many conclusions before the data are presented. Although it is common to have a sentence or two at the end of the Introduction that prepares the reader for the following information, this introduction is really a summary of this paper. Then background material normally presented in an Introduction section is mixed in with results – this makes it less clear what the new information is here.
3. Line 97, change sentence: “the respective Ub variant is likewise differently” to ...”functionally different”.
4. Figure 1b shows the activity of ubiquitin on various lysine derivatives. This experiment is identical to one presented in

reference 11 with the addition of two new lysine derivatives that are the focus of this paper. The time course data between figure 1B and reference 11 figure 1e are quite different with bands disappearing at different rates. Can the authors comment on the reproducibility of this assay or provide an explanation for these differences?

5. Figure 1 also has a typo: line 759, autoubiquitination is misspelled. Also, on line 761, the phrase: "performed as described in Methods and " is not needed and can be removed.

6. Figure 2 – please mark position 11 on one of the structures.

7. Line 315 change "While" to "Whereas" since "While" requires two simultaneous situations.

8. Line 349 – D₂O signals are not typically measured directly as a reference in the spectrum – the instrument locks onto the deuterium signal and then this is used as a reference frequency for other nuclei. Suggested revision: Deuterium lock signal was used as a frequency reference.

9. Line 437 – 5% D₂O is used as a lock reference frequency. Spin locking is usually a term that refers to maintaining the magnetization along a specific axis. Deuterium locking samples the deuterium signal rapidly but does not impose phase coherence along a specific axis.

Reviewer #2

(Remarks to the Author)

Kienle et al. report studies aimed at establishing ketolysine (KeK) as an acetyllysine analogue suitable for use in cellular studies of lysine acetylation function. KeK has one key advantage, namely that it cannot be processed by KDACs and should therefore remain stable after incorporation into a target protein. However, the data provided to indicate that KeK might be a functional mimetic of AcK are either indirect and incomplete, or show instead that it doesn't adequately mimic the function of AcK.

The first part of the paper builds upon the Nature Communications paper published by the same group in 2022. In the earlier paper, they find that acetylation of ubiquitin (Ub) at K11 (AcK) causes a characteristic distortion of the Ub structure that is different from that caused by acetylation of other lysines in Ub. In addition, they find that other K11 mutations, for example to A or Q, do not cause the same structural changes and that AcK11 is a less competent substrate in an ubiquitination assay (judged by Ub consumption) than WT, K11Q or K11A. In the current manuscript, the authors add additional analogues KeK and TFAcK (trifluoroacetyl lysine), finding that these substitutions for K11 cause structural distortions similar to those of AcK (as probed by NMR), and that these new Ub proteins are similarly poor substrates in the ubiquitination assay. These data alone don't provide any positive evidence that KeK is a functional mimic of AcK since the main readout is lack of function.

In the next major section, they conduct pulldown/mass spec experiments to compare the various Ub K11 mutants ability to interact with cellular proteins (lysates). In this experiment that data are more positive, in that there are some proteins that apparently interact preferentially with AcK, KeK, and TFAcK Ub relative to the other probed mutants, as well as some proteins that bind less well to AcK and its candidate mimetics. The cluster and correlation analyses presented appear convincing and show similarity of K11/R11 interacting proteins to each other, Q11/A11 to one another, and AcK/TFAcK/KeK to one another. A handful of examples is validated by pulldown/western in Fig. S3, and the results are generally consistent with the results of the MS study. The weakness of this set of experiments is that there is no indication that any of the interactions has any biological function much less one that is known (or hypothesized) to be acetylation dependent (the point is completely undiscussed), and so again there is no test of whether KeK is functionally mimicking AcK. The idea that the Kac-like mutants are partially destabilized and therefore binding to a collection of proteins non-specifically would seem to be a possible explanation of the data.

The final section of the paper attempts to address the shortcomings of the Ub K11 model system by introducing the same lysine substitutions into known sites of lysine acetylation in the p53 DNA-binding domain. This is a promising choice, since there are positive transcriptional readouts that have been ascribed to acetylation at individual sites, for example *PUMA* induction upon K120 acetylation. However this example, in particular, undercuts the hypothesis that KeK is a AcK mimetic, as there is very little *PUMA* induction compared to TFAcK which should be at least partially modified. (Do we expect in this experiment the same acetylation status as in Fig 5c? Was it determined for each experiment?) To aid in interpretation, it would have been nice to see K120R and K120Q added to these Supplemental experiments where a positive readout is expected. The conclusion would seem to be that p53 K120KeK doesn't produce the expected effect (based on literature), but it's different from AcK (mostly deacetylated in the cell anyway), so it actually might mimic AcK.

Likewise, interpreting the p21 induction experiment of Fig 5 to say that the lack of induction (here, expected for Kac based on literature) shows that KeK is a good AcK mimetic seems too much to me. I think it would be necessary to also show that a stimulus that increases acetylation of K120 in WT (or the addition of KDAC inhibitors to the AcK mutant cells) produces a similar suppression of p21 and does not suppress p21 in the K120R or K120Q mutant case. In essence, there is no experiment in the present study that benchmarks what the effect of p53 Kac is relative to unmodified lysine. Without this, I don't think one can reach any conclusion about whether KeK does or does not mimic Kac functionally.

A glaring issue the authors don't discuss at all is whether Kac reader proteins can recognize KeK mutant forms. These reader modules generally don't bind to K-to-Q mutants, and this lack of recognition aligns with the authors' contention that there is more to Kac than charge neutralization. The Kac readers are especially relevant to the p53 case, where there are reports that Kac at certain sites helps recruit transcriptional activators via reader domains leading to functional consequences. The case that KeK is a good mimic would be strengthened by introducing it into one of the p53 CTD sites known to be recognized by a bromodomain to see whether the interaction is equally strong. My guess is that it would be seriously impaired. Both bromodomains and YEATS domains show quite pronounced preference for the acetamide group of

Kac, and the lack of the amide in KeK would likely have an impact. In either case, it would be important information for anyone considering using this mimetic in cellular studies where readers might play a role.

Reviewer #3

(Remarks to the Author)

Lysine acetylation plays a central role in a range of biological processes, but the deep exploration of lysine acetylation is hampered by the lack of a method to access homogeneous modified proteins. The incorporation of AcK with GCE provides a nice strategy to obtain the acetylated protein for functional study. The work of Huang et al. reported that the non-hydrolyzable AcK analog ketolysine (KeK) is a suitable mimic for the functional study of protein lysine acetylation (ref 15). This manuscript showed that site-specific incorporation of KeK into ubiquitin closely resembled the structural and functional effects of AcK incorporation. The transcriptional activity of KeK-modified p53 was lost, while hydrolyzable AcK-modified p53 regained wt-like activity. Together, this work indicated that KeK was the most suitable AcK analog to study the effect of lysine acetylation of a POI in vitro and in vivo. I cannot support the publication of this manuscript in its current version due to the following reasons:

1. The non-hydrolyzable acetyllysine analogs and aminoacyl-tRNA synthetase/ tRNA pairs used in this work have been well developed before. Therefore, the novelty and general applicability are significantly compromised.
2. The title of this manuscript is "Non-hydrolyzable acetyllysine analogs to study protein acetylation in vitro and in cells." It would be necessary to study a specific protein deeply to explore significant biological questions after evaluating and proving the availability of a KeK substitute strategy.
3. Other concerns: 1) The autoubiquitination level of HDM2 in Figure 1b should be quantified carefully, especially the confusing lane of 11R. 2) As Figure 2b shows, there are many proteins with remarkably different enrichment between Ub variant proteomics, especially between 11AcK and 11KeK. It would be nice to analyze the function of the above proteins and find some clues for studying lysine acetylation in vitro and in vivo. 3) Some weak interactions might be missed during Co-IP with Ub variants, along with the introduction of some indirect protein interactome. Cross-linking mass spectrometry could be taken into consideration. 4) As protein PTMs are highly dynamic, the ability to maintain acetylation of target protein given by KeK may be harmful for several biological processes. It would be nice to evaluate the cell cytotoxicity of the strategy.

Reviewer #4

(Remarks to the Author)

In this study, Kienel et al. present strong evidence that keto-modified lysine is a suitable analog for functional analysis of lysine acetylation. Based on their previous study (Nature Communications, 2022), in addition to charge neutralization, steric effect also contributes to the acetylation-induced ubiquitin structure changes. On this basis, the authors incorporated two acetylated-lysine analogs (Trifluoro-acetyl-lysine(TFAcK) and keto-lysine(KeK)) into target position of ubiquitin (K11) with generic code expansion technology. By showing their similar behaviors to AcK in the E3 ligase HDM2 autoubiquitination assay and the nuclear magnetic resonance spectroscopy, TFAcK and KeK analogs successfully mimicked the properties of acetylation at K11 in ubiquitin, in stark contrast to the traditional glutamine replacement. The authors also investigated the interactome of acetylated ubiquitin in whole extract of HEK293T cells by pull-down mass spectrometry experiment, and compared the differences in binding preference of interact proteins across all their analog candidates, including Ub11 R, Q, A, TFAcK, KeK and other Ub variants (K, AcK, empty control). With hierarchical clustering of interactors and correlation calculation based on MS quantification results, it was found that two candidates (TFAcK and KeK) performed most similar to AcK. Finally, they successfully incorporated the analogs into p53 in human cells, and analyzed their effects on transactivation function of p53. With the observation of "abnormal" less active performance of KeK-p53 variant and the evidences of PRM-MS data, the authors also proved that KeK is much less likely to be removed by deacetylases than AcK and other ones, suggesting it is a more suitable non-hydrolyzable analog for the acetylation functional analysis. The study of the biological function of acetylation on ubiquitin is still very lacking, and this work proposed a promising shortcut to achieve this. Overall, the data are solid and the work is clearly written, thus is worth to be published.

Specific comments:

1. Among the 338 significant interact proteins in the HEK293T extract, in addition to cluster 1 protein list which contains similar interacts between Ub11 AcK, TFAcK and KeK, there is still a large proportion of proteins that exhibit obvious differences in the quantification results between the three analogs. It is recommended to mention this difference and discuss possible reasons.
2. Although the authors performed PRM MS experiment to demonstrate the non-hydrolyzable property of KeK and analyzed its chemical differences with TFAcK during deacetylation. However, the data at this part is somewhat thin. Time-resolved PRM MS analysis for the deacetylation of analogs with related enzymes can further highlight KeK's non-hydrolyzable property and make the result more convincing.
3. Since the authors have mentioned that a combination of PTMs seems to be required for biological functions, such as transcriptional effects in the discussion. It is also necessary to discuss the possible drawbacks of this KeK analog in acetylation studies, such as the potential effects on cross-talk with other modifications, especially for its non-hydrolyzable property.

Version 1:

Reviewer comments:

Reviewer #1

(Remarks to the Author)

The authors have addressed my comments fully.

Reviewer #2

(Remarks to the Author)

The revised manuscript of Keinle, et al. has addressed several points raised in my earlier review. Specifically, the authors have added discussion of the possible role of acetyl lysine reader domains in assessing the functional mimicry potential of KeK, and have included new data comparing p53 K120Q and K120R to other variants (in sup. Figure 6c).

However, other questions remain unanswered. I appreciate that the authors tried to assess the interaction of example bromodomains with the p53 CTD KeK variant, but given the technical difficulties they describe in the response letter, I wonder why they did not instead explore this question using synthetic peptides (they synthesized KeK, so I assume they could synthesize a protected variant for SPPS). There is no compelling reason why the model system must be p53 here—the relevant point is to directly compare AcK to KeK. I think the authors are missing something important by concluding from cursory inspection of bromodomain co-crystal structures that the NH group of the acetamide isn't really doing anything. The noted water-mediated hydrogen bonds are highly conserved positionally and in some cases the relevant water is substituted by the the carbonyl oxygen of a second AcK in the same bound ligand (contributing to the higher affinity observed for diacetyl peptides binding to some bromodomains). As the authors rightly observe, there is an explicit role for the NH group in YEATS domain interactions, and this would not be recapitulated in KeK. Finally, in addition to the NH itself, the electronic properties of amide and ketone oxygens are not the same, potentially affecting the interactions. It may be the case that these readers play no role in the specific experiments reported in this manuscript, but any argument that KeK is a GENERAL and SUPERIOR AcK mimetic is compromised by not including clear experiments comparing binding of these to representative reader domains.

A final point on loss of function: this can of course be of biological importance, but the argument is not philosophical. It remains the case that the KeK experiments shown here that purport to show functional mimicry in a biologically relevant model system reach this conclusion based on LACK of function. Surely the authors would not go so far as to assert that ANY lysine mutation that disrupts a protein function is an acetyl lysine structural mimetic. The only positive mimicry data relate to the Ub system (the structural perturbation observed by NMR and the interactome comparison), and it isn't clear whether the Ub effect is biologically relevant.

Reviewer #3

(Remarks to the Author)

Thanks for the authors' detailed response to the initial review comments and efforts in addressing some of the minor points. However, after carefully evaluating the authors' rebuttal and the revised manuscript, I regret to state that my fundamental concerns regarding the functional validation and general applicability of the KeK strategy remain largely unresolved. The study, in its current form, still falls short of providing compelling evidence to support its central claim that KeK is functionally reliable, non-hydrolyzable acetyllysine mimic for studying lysine acetylation. Below, I elaborate on the key issues that prevent me from supporting publication:

1. The genetic incorporation of KeK has been reported before (ref 16) and the authors in ref 16 have suggested that KeK is an unhydrolysable analogue of AcK and suggested its application in p53. "Moreover, KetoK is an unhydrolysable analogue of AcK that represents one of the most important posttranslationally modified amino acids in eukaryotic cells. Since protein acetylation is a reversible process catalyzed by enzymes like histone acetyltransferases and histone deacetyltransferases, direct incorporation of KetoK into proteins in vivo at sites with naturally occurring lysine acetylation will permanently install an AcK mimic and may provide an efficient way to decipher the regulation roles of acetylation in histones, p53 and other transcription regulatory proteins." The authors stated that the Methanomethylophilus alvus synthetase has not been previously used for KeK incorporation. While this is a technical extension (active sites transfer), it does not constitute a conceptual advance.

2. I agree that the central claim is that KeK is a functionally reliable, non-hydrolyzable acetyllysine mimic to study protein acetylation in vitro and in cells. This is a broad claim that necessitates validation across a diverse range of biological processes. However, the authors fail to perform these key experiments in response to both reviewer #2 and #3. The study still relies heavily on structural similarity (NMR) and loss-of-function phenotypes. The current evidence, limited to only two proteins (Ub and p53), is not only insufficient in scope but also problematic in its interpretation, thereby failing to support this claim of generality.

1) In the Ubiquitin model: The primary functional readout (HDM2 autoubiquitination) only shows that KeK, like AcK, leads to a loss of function. This does not constitute positive evidence of functional mimicry. A true functional mimic should be able to recapitulate a gain of function or a specific switched function induced by acetylation, which is not shown here.

2) My previous comment emphasized the necessity of evaluating the strategy's reliability and delving into deep biological questions. The authors' response misunderstands this point. I am not asking for a complete biological story on p53; rather, I am requesting evidence of the strategy's general applicability. This could be achieved by applying the KeK incorporation to a wider range of proteins or known acetylation-dependent processes, even at a preliminary level. Demonstrating that KeK can reliably mimic AcK in multiple, distinct systems would significantly strengthen the methodological foundation of the paper.

3. HDM2 autoubiquitination quantification (Fig. 1b): A scientifically rigorous approach would involve using software like ImageJ to quantify the autoubiquitination level of HDM2 directly or consumption of free ubiquitin at different time points. This would allow for a direct comparison of ubiquitination kinetics between variants. The current reliance on visual inspection of

Coomassie-stained gels is insufficient for a high-impact journal.

Figure presentation: I noticed the better image had been revised. However, the only adjustment of image brightness/contrast for the 11R lane, rather than ensuring equal protein loading across all samples, is not scientifically rigorous.

4. The authors correctly revised the high correlation and overlap between Ub 11AcK and Ub 11KeK interactomes. However, they dismiss the ~23% of differentially enriched proteins as likely technical artifacts from biological replicates, indirect interactome and MS methods. The conclusion remains further investigation.

1) A more thorough analysis should involve a functional enrichment analysis (e.g., GO, KEGG) of the proteins that are unique to Ub 11AcK or Ub 11KeK. Are these proteins enriched for specific pathways or functions? This could provide crucial clues about the subtle functional differences between AcK and its mimic, which is essential for the central claim that KeK is a functionally reliable, non-hydrolyzable acetyllysine mimic to study protein acetylation.

2) While cross-linking MS could offer higher confidence in identifying direct interactors, a compelling alternative to strengthen their findings would be to benchmark their identified interactors against previously published ubiquitin interactomes. Demonstrating that their AE-MS method recovers known, functionally relevant Ub-binding proteins would validate the reliability of their dataset and, by extension, the similarity between AcK and KeK.

5. The authors argued that assessing the cytotoxicity of stable KeK incorporation is challenging. However, an initial evaluation below is both feasible and critical. Transient transfection of the KeK-incorporation system (including proper controls) coupled with a standard cell viability assay (e.g., MTT, CellTiter-Glo) would provide invaluable data. The goal is not to prove the method is non-toxic, but to transparently quantify its potential impact. Understanding the baseline cytotoxicity is essential for future users to appropriately design their experiments and interpret results. A method's value is not diminished by its limitations, but by a failure to characterize them.

Reviewer #4

(Remarks to the Author)

The authors have addressed all my previous concerns. I suggest the publication of the work as is.

Version 2:

Reviewer comments:

Reviewer #2

(Remarks to the Author)

I believe that with the qualifying statement the authors have added to the manuscript (p13), the overall conclusion is less likely to mislead readers. Personally, I would stop short of calling KeK the 'mimic of choice' for studies in cells (until there is broader functional validation), but it isn't my manuscript. I have no further suggestions to make.

Reviewer #3

(Remarks to the Author)

the paper is ready to publish

Point-by-point response to reviewers' comments on " Non-hydrolyzable acetyllysine analogs to study protein acetylation in vitro and in cells " (NCOMMS-24-65817-T)

Reviewer 1

The manuscript: Non-hydrolyzable acetyllysine analogs to study protein acetylation in vitro and in cells presents a very nice study of a lysine substitute that mimics acetylation using a unnatural amino acids, one of which is chemically resistant to deacetylation. This is a significant improvement in the existing strategy to study acetylated lysine and I expect that this chemically resistant derivative will become a significant and important tool in molecular biology with many broad applications.

Two protein host systems are used to test the synthetic acetylated lysine derivatives: Ubiquitin and p53 – both extremely relevant proteins in cell biology. They use the now standard strategy of stop codon inserted into the gene with complementary tRNAs charged with the unnatural lysine to incorporate these lysine derivatives. Two new derivatives include a lysine with a fluorinated acetyl methyl group TFAcK and another missing the terminal side chain nitrogen that normally forms an amide bond with the acetyl group, KeK. The side chain nitrogen makes the attachment of the acetyl group chemically labile; removing this group was a very smart idea. The experimental strategies employed here are designed to reveal significant information. For example, NMR chemical shift mapping is exquisitely sensitive to small changes in structure that might be induced by amino acid changes. The authors asked the question of whether the lysine derivatives affected protein-protein interactions of Ubiquitin in (Figure 3). This is an excellent assay for functional consequences in comparing acetylation mimics.

Their study of p53 reveal the usefulness of the KeK derivative as well as limitations of their TFAcK version. Problems with TFAcK include poor incorporation into p53 and a higher propensity for deacetylation compared to KeK (which appears quite stable).

Thus, we have a winner! KeK appears to be a superior substitution to study acetylated lysine.

In addition to demonstrating the usefulness of the KeK variant, their results show clearly that glutamine is an insufficient mimic of a neutralized (acetylated) lysine. This was proven by independent assays in both the model systems of study (Ubiquitin and p53). These data should serve as cautionary evidence for those researchers exploring the function of modified lysines: be very careful when substituting in glutamine.

*The paper is well-written, the methods are detailed, and the strategy is sound.
(Very) minor suggestions for correction:*

1. The terminology for amino acids varies between single letter codes and spelling out the amino acid name. It would be helpful to use consistent terms.

Thank you for this suggestion. In the revised manuscript, we have attempted to use the single letter code throughout; however, we feel that in certain instances (e.g. when talking about amino acid residues in general), spelling out the full name of an amino acid makes the text more easily readable.

2. Organization: The last paragraph of the Introduction contains many conclusions before the data are presented. Although it is common to have a sentence or two at the end of the Introduction that

prepares the reader for the following information, this introduction is really a summary of this paper. Then background material normally presented in an Introduction section is mixed in with results – this makes it less clear what the new information is here.

Thank you for pointing this out. We have reorganized the last paragraph accordingly.

3. Line 97, change sentence: “the respective Ub variant is likewise differently” to ... “functionally different”.

The sentence has been changed accordingly.

4. Figure 1b shows the activity of ubiquitin on various lysine derivatives. This experiment is identical to one presented in reference 11 with the addition of two new lysine derivatives that are the focus of this paper. The time course data between figure 1B and reference 11 figure 1e are quite different with bands disappearing at different rates. Can the authors comment on the reproducibility of this assay or provide an explanation for these differences?

In our experience, HDM2 is not an "easy-to-handle" protein (to say the least) and, thus, the actual activity of HDM2 differs from protein prep to protein prep. Comparing the time course data between the figures mentioned, it appears that the HDM2 prep used in the present manuscript was somewhat more active than the one used in reference 11. That is, Ub 11AcK was quantitatively consumed by HDM2 in figure 1B at 120 min, whereas in ref 11 figure 1e approximately 50 percent were still present. However, only little auto-ubiquitination is observed at 60 min in both experiments. Similarly, Ub 11Q is more efficiently used by HDM2 in figure 1B at 60 min than at the same time in ref 11 figure 1e; yet in both experiments, it is completely consumed at the later point. Thus, although the data differ between the two figures in a quantitative manner, they are qualitatively similar (i.e. Ub 11AcK is less efficiently used by HDM2 than Ub 11Q).

5. Figure 1 also has a typo: line 759, autoubiquitination is misspelled. Also, on line 761, the phrase: “performed as described in Methods and “ is not needed and can be removed.

The typo has been corrected and the phrase has been removed.

6. Figure 2 – please mark position 11 on one of the structures.

Position 11 has been marked in panels a-c.

7. Line 315 change “While” to “Whereas” since “While” requires two simultaneous situations.

Thank you for indicating this.

8. Line 349 – D2O signals are not typically measured directly as a reference in the spectrum – the instrument locks onto the deuterium signal and then this is used as a reference frequency for other nuclei. Suggested revision: Deuterium lock signal was used as a frequency reference.

We apologize for not having phrased this properly in the first place. The respective sentence has been fixed.

9. Line 437 – 5% D2O is used as a lock reference frequency. Spin locking is usually a term that refers to maintaining the magnetization along a specific axis. Deuterium locking samples the deuterium signal rapidly but does not impose phase coherence along a specific axis.

Thank you for spotting this error. The respective sentence has been corrected.

Reviewer 2

Kienle et al. report studies aimed at establishing ketolysine (KeK) as an acetyllysine analogue suitable for use in cellular studies of lysine acetylation function. KeK has one key advantage, namely that it cannot be processed by KDACs and should therefore remain stable after incorporation into a target protein. However, the data provided to indicate that KeK might be a functional mimetic of AcK are either indirect and incomplete, or show instead that it doesn't adequately mimic the function of AcK.

1. The first part of the paper builds upon the Nature Communications paper published by the same group in 2022. In the earlier paper, they find that acetylation of ubiquitin (Ub) at K11 (AcK) causes a characteristic distortion of the Ub structure that is different from that caused by acetylation of other lysines in Ub. In addition, they find that other K11 mutations, for example to A or Q, do not cause the same structural changes and that AcK11 is a less competent substrate in an ubiquitination assay (judged by Ub consumption) than WT, K11Q or K11A. In the current manuscript, the authors add additional analogues KeK and TFAcK (trifluoroacetyl lysine), finding that these substitutions for K11 cause structural distortions similar to those of AcK (as probed by NMR), and that these new Ub proteins are similarly poor substrates in the ubiquitination assay. These data alone don't provide any positive evidence that KeK is a functional mimic of AcK since the main readout is lack of function.

We agree with the reviewer that the ubiquitination assays do not provide positive evidence that KeK is a functional mimic of AcK, and in fact, we intentionally did not make such a conclusion. As indicated in the manuscript (lines 123-126), the reason for doing these assays was to provide evidence that unlike Q, KeK and TFAcK properly mimic the effects of K11 acetylation on Ub structure. Nonetheless, we respectfully disagree with the notion that loss of function is generally not a functional readout (as loss of function of a protein is in many cases of significant physiological relevance).

2. In the next major section, they conduct pulldown/mass spec experiments to compare the various Ub K11 mutants ability to interact with cellular proteins (lysates). In this experiment that data are more positive, in that there are some proteins that apparently interact preferentially with AcK, KeK, and TFAcK Ub relative to the other probed mutants, as well as some proteins that bind less well to AcK and its candidate mimetics. The cluster and correlation analyses presented appear convincing and show similarity of K11/R11 interacting proteins to each other, Q11/A11 to one another, and AcK/TFAcK/KeK to one another. A handful of examples is validated by pulldown/western in Fig. S3, and the results are generally consistent with the results of the MS study. The weakness of this set of experiments is that there is no indication that any of the interactions has any biological function much less one that is known (or hypothesized) to be acetylation dependent (the point is completely undiscussed), and so again there is no test of whether KeK is functionally mimicking AcK. The idea that the Kac-like mutants are partially destabilized and therefore binding to a collection of proteins non-specifically would seem to be a possible explanation of the data.

We agree that at this stage, we can only speculate about the physiological function of the interactions identified (as we have discussed in ref. 11 of the manuscript). Respective studies would have to be performed in cells and, as indicated in the manuscript (lines 171-172), would be rather challenging for ubiquitin (as it is one of the most highly abundant proteins in eukaryotic cells). Yet, the intention of the AE-MS experiments, and as also indicated by the reviewer, was to provide evidence that incorporation of AcK, KeK, and TFAcK have similar effects on the interaction properties of ubiquitin and, thus, are functionally similar in this aspect. We hope that the reviewer agrees that the ability or property of a protein to interact with other proteins or biomolecules represents the basis for the function of a protein and, thus, is of functional relevance. If so, our data indicate that in this case, KeK functionally mimics AcK. Again, which of the interactions identified are physiologically relevant, remains to be proven.

The idea of the reviewer that the interactome data are possibly explained by non-specific interactions appears highly unlikely to us. This interpretation may hold true for some of the interactions. However, if this (non-specific interactions) were generally the case, we would expect that proteins, such as chaperones, recognizing structurally distorted proteins would be among the interactors, which is not the case. Similarly, the "interactomes" would likely differ from experiment to experiment in a significant manner. In other words, the AE-MS data would not be reproducible, however, they clearly are (as also indicated by the validation experiment with AcK-specific antibodies shown in Supplementary Figure 3).

3. The final section of the paper attempts to address the shortcomings of the Ub K11 model system by introducing the same lysine substitutions into known sites of lysine acetylation in the p53 DNA-binding domain. This is a promising choice, since there are positive transcriptional readouts that have been ascribed to acetylation at individual sites, for example PUMA induction upon K120 acetylation. However this example, in particular, undercuts the hypothesis that KeK is a AcK mimetic, as there is very little PUMA induction compared to TFAcK which should be at least partially modified. (Do we expect in this experiment the same acetylation status as in Fig 5c? Was it determined for each experiment?) To aid in interpretation, it would have been nice to see K120R and K120Q added to these Supplemental experiments where a positive readout is expected. The conclusion would seem to be that p53 K120KeK doesn't produce the expected effect (based on literature), but it's different from AcK (mostly deacetylated in the cell anyway), so it actually might mimic AcK.

The acetylation status was not determined for each experiment (this is technically not feasible, due to the amounts of cell extract required for PRM analysis) and, as shown in Figure 5c, varies for TFAcK from experiment to experiment approximately between 40-70 percent. A similar variation is seen for the transactivation potential of p53-120TFAcK (Figure 5a). In our experience, such variations are inherent to transfection assays performed at different days (and, thus, with cells that for instance, differ in their actual proliferation status).

We are grateful to the reviewer for the suggestion to repeat the Puma reporter experiments with p53-120Q and p53-120R (new Supplementary Figure 6c). The results obtained again show that p53-120TFAcK is more proficient in inducing the expression of the Puma reporter construct than p53-120KeK, but less proficient than wt p53. In contrast to the results obtained with the p21 reporter construct (Figure 5d), p53-120Q is much less efficient in inducing the Puma reporter construct, while p53-120R induces the expression of both reporter constructs with similar efficiency. In other words, p53-120Q behaves more similar to p53-120KeK in the Puma reporter assay. As already indicated in the original manuscript, we are aware that these data are in apparent contrast to the current assumption on the effect of K120 acetylation on PUMA expression. Accordingly, we have extended the discussion on this issue in the revised manuscript

(Discussion, last paragraph, lines 347-364). Nonetheless, our data obtained for the Puma reporter construct are in themselves fully consistent with KeK being a functional mimic of AcK, as p53-120KeK displays ~10-30 percent activity compared to wt p53 and the activity of p53-120TFAcK varies between ~60-80 percent of wt p53.

4. Likewise, interpreting the p21 induction experiment of Fig 5 to say that the lack of induction (here, expected for Kac based on literature) shows that KeK is a good AcK mimetic seems too much to me. I think it would be necessary to also show that a stimulus that increases acetylation of K120 in WT (or the addition of KDAC inhibitors to the AcK mutant cells) produces a similar suppression of p21 and does not suppress p21 in the K120R or K120Q mutant case. In essence, there is no experiment in the present study that benchmarks what the effect of p53 Kac is relative to unmodified lysine. Without this, I don't think one can reach any conclusion about whether KeK does or does not mimic Kac functionally.

While we appreciate the suggestion of the reviewer, we deliberately refrained from doing such experiments. It is commonly accepted that acetylation plays a general role in transcriptional transactivation and, thus, interfering with the activity of acetylases or deacetylases has rather pleiotropic effects, making interpretation of respective results at least difficult. For instance, Tip60 is one of the enzymes that acetylates p53 at K120; yet, according to literature, it not only stimulates the expression of PUMA and BAX (Sykes SM et al., Mol Cell 24, pp. 841; Tang Y et al., Mol Cell 24, pp. 827), it also stimulates p21 expression (Doyon Y et al., Mol Cell Biol 24, pp. 1884), which supposedly occurs in the absence of p53 acetylation (Sykes et al).

5. A glaring issue the authors don't discuss at all is whether Kac reader proteins can recognize KeK mutant forms. These reader modules generally don't bind to K-to-Q mutants, and this lack of recognition aligns with the authors' contention that there is more to Kac than charge neutralization. The Kac readers are especially relevant to the p53 case, where there are reports that Kac at certain sites helps recruit transcriptional activators via reader domains leading to functional consequences. The case that KeK is a good mimic would be strengthened by introducing it into one of the p53 CTD sites known to be recognized by a bromodomain to see whether the interaction is equally strong. My guess is that it would be seriously impaired. Both bromodomains and YEATS domains show quite pronounced preference for the acetamide group of Kac, and the lack of the amide in KeK would likely have an impact. In either case, it would be important information for anyone considering using this mimetic in cellular studies where readers might play a role.

Thank you for this suggestion. Using a bacterial expression system, we have generated a CTD variant (amino acids 325-393) with AcK at position 382 (CTD-382AcK) and tested its interaction with two bromodomains, one of which was derived from p300 (a known interactor of p53), in conventional pulldown experiments, by spectral shift analysis (Monolith X, Nanotemper), and by structural mass spectrometry (i.e. native mass spectrometry) (the latter two allow the detection of low affinity interactions). Unfortunately, we could not obtain reliable results with any of the methods. We believe that this is because of two main reasons. Firstly, the CTD of p53 forms mostly tetramers. Despite considerable efforts, we did not manage to obtain CTD-382AcK preparations that contained more than 50 percent of the "full-length" protein. That is, our CTD preparations consisted of 50 percent CTD-382AcK and 50 percent of a respectively truncated version (amino acids 325-381). Thus, we assume that the interaction of the respective tetramers with the bromodomains used was too weak to be detected. Secondly, when increasing the concentrations of CTD-382AcK and/or the bromodomain in our interaction assays, we observed

aggregate formation. It is therefore likely that we were not able to reach the concentrations required to reliably detect low affinity interactions between the CTD of p53 and bromodomains.

We were somewhat puzzled by the notion of the reviewer that "bromodomains and YEATS domains show quite pronounced preference for the acetamide group of Kac". According to published structures, in addition to hydrophobic effects, the most important contact is a direct hydrogen bond between distinct residues of the AcK binding domains and the carbonyl group. For YEATS domains, there is an additional direct hydrogen bond with the nitrogen of the amide bond, and mutation of the respective residue has a significant effect on the K_d of the interaction (though not as significant as mutation of the residue contacting the carbonyl group) (e.g. Li et al., Cell 159, pp.558). For bromodomains, the situation seems different inasmuch as there does not appear to be a direct hydrogen bond between the bromodomain and the nitrogen of the amide bond; both are rather involved in hydrogen bonding of water molecules located at the interface (e.g. Owen et al., EMBO J 19, pp. 6141). Thus, based on the high structural similarity of AcK and KeK (as also indicated by high resolution NMR spectroscopy with ubiquitin), we would not expect that substitution of the nitrogen by carbon has a dramatic impact on the interaction with bromodomains.

Since we fully agree with the reviewer that readers (i.e. interacting proteins) are important for the function of acetylated proteins, we have incorporated a respective discussion in the revised manuscript (lines 328-346), with the conclusions that (1) in our transactivation assays, bromodomain-containing proteins or other AcK binding proteins do not appear to play a critical role (we are aware that this does not show that KeK is recognized by AcK binding proteins with an affinity similar to AcK), and (2) a comparative analysis of variants of a protein of interest (POI), in which a distinct K residue is replaced by R, Q, or AcK surrogates, provides critical information about if and how acetylation affects known properties/functions of a POI.

Reviewer 3

Lysine acetylation plays a central role in a range of biological processes, but the deep exploration of lysine acetylation is hampered by the lack of a method to access homogeneous modified proteins. The incorporation of AcK with GCE provides a nice strategy to obtain the acetylated protein for functional study. The work of Huang et al. reported that the non-hydrolyzable AcK analog ketolysine (KeK) is a suitable mimic for the functional study of protein lysine acetylation (ref 15). This manuscript showed that site-specific incorporation of KeK into ubiquitin closely resembled the structural and functional effects of AcK incorporation. The transcriptional activity of KeK-modified p53 was lost, while hydrolyzable AcK-modified p53 regained wt-like activity. Together, this work indicated that KeK was the most suitable AcK analog to study the effect of lysine acetylation of a POI in vitro and in vivo. I cannot support the publication of this manuscript in its current version due to the following reasons:

We respectfully disagree with the statement "*The work of Huang et al. reported that the non-hydrolyzable AcK analog ketolysine (KeK) is a suitable mimic for the functional study of protein lysine acetylation.*" Huang et al. nicely show that KeK can be incorporated into GFP in bacteria, but do not provide any experimental evidence that it is a suitable mimic for studying functional aspects of protein acetylation (they simply mention this possibility in the last sentence of the manuscript).

1. The non-hydrolyzable acetyllysine analogs and aminoacyl-tRNA synthetase/ tRNA pairs used in this work have been well developed before. Therefore, the novelty and general applicability are significantly compromised.

We agree that the analogs and the synthetase/tRNA pairs used in our work have been developed before and, thus, are not novel per se (respective credit was given in the original manuscript, cf. ref. 16 of the revised manuscript). Yet, we respectfully disagree with the notion that the general novelty and applicability is therefore significantly compromised. Here we show for the first time that the synthetase/tRNA pair derived from *Methanomethylophilus alvus* (please note that this synthetase has not been used before for incorporation of KeK) and the pair derived from *Methanosarcina mazei* can be used to introduce AcK, TFAcK, or KeK at defined positions into a protein of interest in bacteria (ubiquitin, p53) and in mammalian cell lines (p53), respectively, thereby enabling subsequent functional analyses.

2. The title of this manuscript is "Non-hydrolyzable acetyllysine analogs to study protein acetylation in vitro and in cells." It would be necessary to study a specific protein deeply to explore significant biological questions after evaluating and proving the availability of a KeK substitute strategy.

We interpret the comment of the reviewer such that she/he agrees that our data indicate that the "KeK substitution strategy" is in general suitable to study protein acetylation. If this interpretation is correct, we would then hope that the reviewer also agrees that our experiments with p53 address a significant biological question. Although they are at an initial stage, they should spark immediate interest in the p53 community and beyond to reevaluate published results concerning the effects of p53 acetylation as well as acetylation of other proteins of interest. However, such studies are, in our opinion, clearly beyond the scope of the present manuscript.

3. Other concerns:

1) The autoubiquitination level of HDM2 in Figure 1b should be quantified carefully, especially the confusing lane of 11R.

We generally monitor autoubiquitination reactions by Coomassie blue staining. The relative efficiency of the reactions can be most easily compared by looking at the levels of free Ub (as indicated in the revised manuscript on page 5, line 110)). We deliberately did not quantify the respective levels, as the assay per se is not quantitative (i.e. relative differences are scored). However, if the reviewer feels this to be important, we will be happy to do so. Concerning "the confusing lane of 11R", we have provided a better image in the revised manuscript. In addition, please see also our response to comment 4 of reviewer 1 and ref. 11 of the manuscript.

2) As Figure 2b shows, there are many proteins with remarkably different enrichment between Ub variant proteomics, especially between 11AcK and 11KeK. It would be nice to analyze the function of the above proteins and find some clues for studying lysine acetylation in vitro and in vivo.

We assume that similar to reviewer 4 (see below, comment 1) is referring to the notion that according to the heatmap shown in Figure 3b, not all of the interactors identified are shared between Ub 11AcK and Ub 11KeK. We have therefore taken another look at our data and compiled a list of the top 100 significantly enriched binders of 11AcK and Ub 11KeK (Supplementary Figure 3d). This revealed that more than three quarters (77%) of these most significant interactions are shared between Ub 11AcK and Ub 11KeK. In combination with our correlation plot (Figure 3c), this overwhelming similarity in detected interactors underscores the similarity of Ub 11AcK and Ub 11KeK and the suitability of using KeK as an AcK surrogate for functional analysis of lysine acetylation.

Regarding the 20-25% of the most enriched interactors that are not shared between Ub 11AcK and Ub 11KeK, this could be due to systemic/technical limitations of the AE-MS approach (for more

details, please see our response to comment 1 of reviewer 4) and/or to the notion that the structure of Ub 11AcK and Ub 11KeK are highly similar but not identical (Figure 2). We therefore feel that taking a closer look at these apparently differently enriched interactors is not warranted at this stage.

3) Some weak interactions might be missed during Co-IP with Ub variants, along with the introduction of some indirect protein interactome. Cross-linking mass spectrometry could be taken into consideration.

We fully agree with the reviewer that cross-linking mass spectrometry (XL-MS) and affinity enrichment MS are complementary approaches and that XL-MS can be used, for example, to substantiate direct binding partners within interactomes obtained by affinity enrichment MS. However, as the sensitivity of XL MS still significantly lags behind, it is very challenging to obtain a comparable comprehensive picture. This is particularly true, where the amount of bait protein is limited, as is the case with some of the Ub variants. We therefore feel that obtaining complementary interactomes by XL-MS is clearly beyond the scope of the current study.

4) As protein PTMs are highly dynamic, the ability to maintain acetylation of target protein given by KeK may be harmful for several biological processes. It would be nice to evaluate the cell cytotoxicity of the strategy.

The reviewer raises an important issue. Depending on the protein of interest, we would expect that maintaining the acetylation status will have cytotoxic consequences, and these will likely be more dramatic for long-lived proteins than for short-lived proteins. Ideally, this will have to be tested in cells stably expressing the synthetase/tRNA pair and the protein of interest. However, this is a rather challenging task and beyond the scope of the manuscript. In transient transfection assays (as performed here), this will be difficult to address, as for instance expression levels cannot be maintained for a longer period. In our case, assessing the potential toxicity of stably acetylated p53 forms is not possible, as it is commonly accepted that ectopic expression of wild-type p53 is cytotoxic per se (which is the reason, why we usually analyze the transactivation potential of p53 24 h upon transfection).

Reviewer 4

In this study, Kienle et al. present strong evidence that keto-modified lysine is a suitable analog for functional analysis of lysine acetylation. Based on their previous study (Nature Communications, 2022), in addition to charge neutralization, steric effect also contributes to the acetylation-induced ubiquitin structure changes. On this basis, the authors incorporated two acetylated-lysine analogs (Trifluoro-acetyl-lysine (TFAcK) and keto-lysine (KeK)) into target position of ubiquitin (K11) with generic code expansion technology. By showing their similar behaviors to AcK in the E3 ligase HDM2 autoubiquitination assay and the nuclear magnetic resonance spectroscopy, TFAcK and KeK analogs successfully mimicked the properties of acetylation at K11 in ubiquitin, in stark contrast to the traditional glutamine replacement. The authors also investigated the interactome of acetylated ubiquitin in whole extract of HEK293T cells by pull-down mass spectrometry experiment, and compared the differences in binding preference of interact proteins across all their analog candidates, including Ub11 R, Q, A, TFAcK, KeK and other Ub variants (K, AcK, empty control). With hierarchical clustering of interactors and correlation calculation based on MS quantification

results, it was found that two candidates (TFAcK and KeK) performed most similar to AcK. Finally, they successfully incorporated the analogs into p53 in human cells, and analyzed their effects on transactivation function of p53. With the observation of “abnormal” less active performance of KeK-p53 variant and the evidences of PRM-MS data, the authors also proved that KeK is much less likely to be removed by deacetylases than AcK and other ones, suggesting it is a more suitable non-hydrolyzable analog for the acetylation functional analysis. The study of the biological function of acetylation on ubiquitin is still very lacking, and this work proposed a promising shortcut to achieve this. Overall, the data are solid and the work is clearly written, thus is worth to be published.

Specific comments:

1. Among the 338 significant interact proteins in the HEK293T extract, in addition to cluster 1 protein list which contains similar interacts between Ub11 AcK, TFAcK and KeK, there is still a large proportion of proteins that exhibit obvious differences in the quantification results between the three analogs. It is recommended to mention this difference and discuss possible reasons.

The reviewer correctly points out that a significant proportion of proteins, which do not belong to any of the clusters currently discussed in the manuscript, are differentially enriched between the three Ub variants and we agree that this should be discussed (see lines 264-268). In this context, it should be noted that proteome-wide affinity enrichments in lysates typically exhibit a certain amount of variance. Slight differences in the growth state of the cells averaged within a biological replicate are only one contributing factor. Also, weak interactions and indirect interaction partners may also slightly vary between replicate measurements. It is also important to point out that current AE-MS approaches, despite impressive technological advancements in sensitivity and speed of available MS-technology over the last two decades, are still likely to under-sample protein interactions on a proteome-wide scale. Variations between identified interactors of Ub variants are therefore expected. In addition, the structures of Ub 11AcK, Ub 11TFAcK, and Ub 11KeK are highly similar but not identical (Figure 2), presumably contributing to the differences in their interaction profiles.

Having said that, in all our experiments/analyses, Ub 11KeK clustered next to Ub 11AcK, strongly suggesting that KeK is indeed the most similar surrogate to AcK. This is not only apparent in our heatmap (Figure 3b), but is also reflected in our correlation plot employing all significantly enriched 338 interactors (Figure 3c). Most importantly, and in line with the suggestion of the reviewer, we have also taken another look at our data and compiled a list of the top 100 significantly enriched binders of 11AcK and Ub 11KeK (Supplementary Figure 3d). This revealed that more than three quarters (77%) of these most significant interactions are shared between Ub 11AcK and Ub 11KeK. In combination with our correlation plot (Figure 3c), this overwhelming similarity in detected interactors underscores the similarity of Ub 11AcK and Ub 11KeK and the suitability of using KeK as an AcK surrogate for functional analysis of lysine acetylation.

2. Although the authors performed PRM MS experiment to demonstrate the non-hydrolyzable property of KeK and analyzed its chemical differences with TFAcK during deacetylation. However, the data at this part is somewhat thin. Time-resolved PRM MS analysis for the deacetylation of analogs with related enzymes can further highlight KeK's non-hydrolyzable property and make the result more convincing.

Thank you for this suggestion. We assume that the reviewer is asking for a time course experiment in vitro with distinct deacetylases. As shown in the *figure for the reviewer*, p53-120AcK was readily

deacetylated by the NAD-dependent deacetylase Sirt-2, while p53-120KeK remained stable. Unfortunately, even p53-120AcK was not deacetylated by the Zn-dependent deacetylase HDAC2 in a reproducible manner. This may indicate that p53-120AcK does not represent a substrate for HDAC2 and/or that other Zn-dependent enzymes mediate deacetylation of p53-120AcK in cells. We can also not exclude that the enzyme used was simply not or only poorly active, as we do not have a positive control.

Figure for the reviewer. **a**, p53-120AcK and p53-120KeK were incubated with Sirt2 (Höllmüller et al., J Proteome Res 20, pp. 4443) in the presence or absence of NAD or with HDAC2 (Reaction Biology Corp.) at 37 °C for the times indicated. Aliquots of the reactions were subjected to SDS-PAGE followed by western blot analysis using an antibody specifically recognizing p53 acetylated at K120 (upper panel) and an antibody (DO1) recognizing p53 in general (lower panel) as loading control. **b**, The Ub 11 variants indicated were incubated with HDAC6 (Active Motif S.A.) at 30 °C for 3 h, and reaction products were analyzed by intact protein mass spectrometry. Peak areas of the respective acetylated and deacetylated forms of the Ub variants were used for quantification and normalized to the total Ub peak area. Data was obtained in two independent experiments.

We previously provided evidence that HDAC6 can act as a deacetylase of acetylated Ub variants (ref. 11 of the manuscript). Thus, we determined the efficiency of HDAC-mediated deacetylation of Ub 11AcK, Ub11 TFAcK, and Ub 11KeK by intact protein MS analysis. This showed that as expected Ub 11AcK and Ub 11TFAcK were deacetylated by HDAC6 but not Ub 11KeK (figure for the reviewer). However, the percentage was rather low (approximately 10-15 percent). We therefore do not feel comfortable to present the data in the manuscript. However, based on the mechanism of HDAC-catalyzed hydrolysis of the amide bond of AcK, it is highly unlikely that HDACs are capable of hydrolyzing the C-C bond in case of KeK-modified proteins. We have added a respective discussion to the revised manuscript and hope that the reviewer agrees that in combination with our cellular data, this makes a strong case that KeK represents a non-hydrolyzable analog of AcK (lines 289-296: *In contrast to TFAcK, KeK is resistant against the attack of Zn-dependent deacetylases as well. An intermediate step in HDAC-catalyzed*

deacetylation is the nucleophilic attack of the carbonyl C atom by a water molecule. To do so, a base, presumably a histidine residue, accepts a proton from the water molecule and transfers it to the nitrogen atom of the amide bond, which facilitates the final cleavage of the amide bond⁴⁰. In contrast to the nitrogen, the carbon atom of KeK (Figure 1a) can most probably not function as an acceptor for the proton. Thus, while in the case of KeK the nucleophilic attack may still occur, though with reduced efficiency, cleavage of the C-C bond is highly unlikely).

3. Since the authors have mentioned that a combination of PTMs seems to be required for biological functions, such as transcriptional effects in the discussion. It is also necessary to discuss the possible drawbacks of this KeK analog in acetylation studies, such as the potential effects on cross-talk with other modifications, especially for its non-hydrolyzable property.

Thank you for this suggestion. We have incorporated a brief discussion along this line in the revised manuscript (lines 358-64: *For instance, it could be envisioned that such phosphorylation events occur only under certain stress conditions. An alternative, but not mutual exclusive possibility is that to be functional, such PTMs have to occur in a timely ordered, i.e. consecutive manner. Along this line, prior lysine acetylation, in particular incorporation of an AcK mimic that cannot be hydrolyzed as in our case, may prevent phosphorylation of p53 at residues required for transcriptional transactivation. To address such possibilities*).

Point-by-point response to reviewers' comments on " *Non-hydrolyzable acetyllysine analogs to study protein acetylation in vitro and in cells* " (NCOMMS-24-65817-A)

Reviewer 2

The revised manuscript of Keinle, et al. has addressed several points raised in my earlier review. Specifically, the authors have added discussion of the possible role of acetyl lysine reader domains in assessing the functional mimicry potential of KeK, and have included new data comparing p53 K120Q and K120R to other variants (in sup. Figure 6c).

However, other questions remain unanswered. I appreciate that the authors tried to assess the interaction of example bromodomains with the p53 CTD KeK variant, but given the technical difficulties they describe in the response letter, I wonder why they did not instead explore this question using synthetic peptides (they synthesized KeK, so I assume they could synthesize a protected variant for SPPS). There is no compelling reason why the model system must be p53 here—the relevant point is to directly compare AcK to KeK. I think the authors are missing something important by concluding from cursory inspection of bromodomain co-crystal structures that the NH group of the acetamide isn't really doing anything. The noted water-mediated hydrogen bonds are highly conserved positionally and in some cases the relevant water is substituted by the the carbonyl oxygen of a second AcK in the same bound ligand (contributing to the higher affinity observed for diacetyl peptides binding to some bromodomains). As the authors rightly observe, there is an explicit role for the NH group in YEATS domain interactions, and this would not be recapitulated in KeK. Finally, in addition to the NH itself, the electronic properties of amide and ketone oxygens are not the same, potentially affecting the interactions. It may be the case that these readers play no role in the specific experiments reported in this manuscript, but any argument that KeK is a GENERAL and SUPERIOR AcK mimetic is compromised by not including clear experiments comparing binding of these to representative reader domains.

Although we have no expertise with respect to solid phase peptide synthesis, we could synthesize a protected KeK variant and cooperate with a respective group for synthesis. However, we deliberately decided not to follow this route.

We agree with the reviewer that the NH group forms a hydrogen bond with a water molecule and that this would not be possible by substitution of the nitrogen by carbon. We are also aware that the +M effect in amides is somehow unique and a well-known feature of such bonds that cannot be fully mimicked by bonds that are not hydrolyzable. Thus, we believe that the ketone surrogate is the best-possible mimic, in particular when considering the keto-enol tautomer that is somewhat comparable to the +M effect in amides insofar as it results in a partial negative charge at the carbonyl oxygen. Accordingly, we did not conclude that the acetamide "isn't really doing anything", we rather concluded "we would not expect that substitution of the nitrogen by carbon has a *dramatic* impact on the interaction with bromodomains". However, we recognize that considering the published data with TFAcK, we should have been more cautious in our response (we hope that the reviewer agrees that in the Discussion of the first revision of the manuscript, we addressed this issue appropriately). Miller et al. (ref. 15 of the manuscript) performed such peptide-based experiments with TFAcK. In microarray studies with 9-mers derived from H3 containing either AcK or TFAcK at the respective position and bromodomains derived from BAZ2B and BRD9, they observed that for some peptides, the interaction with TFAcK-containing peptides was similar to that of AcK-containing peptides, while for other peptides, binding to TFAcK-containing peptides was significantly reduced or not detected. In addition, in ITC studies using 12-mers, the affinity of some of the bromodomains studied for TFAcK-containing peptides was decreased by about 4-8 fold

(which we would call "not dramatic"). The latter observation is potentially due to the electron-withdrawing effect of the trifluoromethyl group.

The results obtained by Miller et al. (i.e. interactions depend on the peptide and the bromodomain used) indicate that even if a given bromodomain interacts with similar affinities with a given KeK-containing peptide and with the respective AcK-containing peptide, it will not be possible to conclude that KeK is a suitable AcK mimetic in all cases. Because of this and the notion that results obtained with short peptides have to generally be interpreted with caution (e.g. because of structural limitations), we decided to study a physiologically relevant interaction related to our system, the interaction of p300 with a properly folded protein or protein domain (i.e. C-terminal region of p53 including its oligomerization domain). This unfortunately did not work out. Yet, even if the data were positive, this would not mean that KeK is a suitable AcK mimetic in all cases. Rather, this will have to be determined for each individual case, as we had indicated in the Discussion of the revised manuscript.

We would also like to point out that we did not argue that "*KeK is a GENERAL and SUPERIOR AcK mimetic*". We rather proposed that "*KeK is the AcK surrogate of choice for studying acetylation of a given protein IN CELLS*", with the limitations indicated in the Discussion of the manuscript (and above). This proposition is based on the structural similarity of AcK and KeK, which is also shared by TFAcK, and the fact that in contrast to AcK and in part TFAcK, KeK cannot be hydrolyzed. As such, KeK is indeed *superior* to TFAcK. In addition, we would argue (for obvious reasons) that there is, and will be, no AcK analog that mimics the properties/features of AcK to 100 percent. Along this line and according to our experience, we would propose to generally use AcK for in vitro studies (as hydrolysis does not appear to be a matter of concern, unless studying the interaction with deacetylases). However, since AcK is efficiently hydrolyzed in cells, we proposed that KeK is currently the most suitable mimic of AcK for studies *in cells*. Yet, as discussed above and in the manuscript, we agree that KeK has its limitations, as it may not be recognized by all interactors of a protein of interest with an affinity similar to AcK. To make this limitation more clear, we added an additional sentence to the Discussion (page 13, lines 345-349: *Because it is not hydrolyzable, KeK represents the AcK mimic of choice for such studies in cells, with the potential limitation that in some cases, protein(s) critically involved in executing the functional consequences of acetylation may recognize KeK less efficiently than AcK. For in vitro studies, we propose to use AcK, as long as hydrolysis is not a matter of concern.*).

A final point on loss of function: this can of course be of biological importance, but the argument is not philosophical. It remains the case that the KeK experiments shown here that purport to show functional mimicry in a biologically relevant model system reach this conclusion based on LACK of function. Surely the authors would not go so far as to assert that ANY lysine mutation that disrupts a protein function is an acetyl lysine structural mimetic. The only positive mimicry data relate to the Ub system (the structural perturbation observed by NMR and the interactome comparison), and it isn't clear whether the Ub effect is biologically relevant.

Needless to say that we agree with the reviewer that not "*ANY lysine mutation that disrupts a protein function is an acetyl lysine structural mimetic*". In fact, although there is no prove of biological relevance, the Ub interactome data clearly show that KeK is a rather good AcK mimic, with the potential limitation discussed above. Moreover, both Ub11AcK and Ub11KeK interact with a set of proteins that do not or not detectably interact with non-modified Ub. That is, Ub11AcK/Ub11KeK have not just *lost* binding properties but have also *gained* binding properties. At least in our opinion, the important conclusion of the data obtained with p53 is that results regarding the potential consequences of acetylation for a given protein in cells that were obtained by indirect means should generally be interpreted with caution. We would also like to emphasize

that our AcK- and KeK-containing p53 variants still have the potential to bind sequence-specifically to DNA, excluding "unspecific" effects caused by structural distortions. Thus, at least in this respect, the p53 variants show no loss of function.

Reviewer 3

Thanks for the authors' detailed response to the initial review comments and efforts in addressing some of the minor points. However, after carefully evaluating the authors' rebuttal and the revised manuscript, I regret to state that my fundamental concerns regarding the functional validation and general applicability of the KeK strategy remain largely unresolved. The study, in its current form, still falls short of providing compelling evidence to support its central claim that KeK is functionally reliable, non-hydrolyzable acetyllysine mimic for studying lysine acetylation. Below, I elaborate on the key issues that prevent me from supporting publication:

1. The genetic incorporation of KeK has been reported before (ref 16) and the authors in ref 16 have suggested that KeK is an unhydrolysable analogue of AcK and suggested its application in p53. "Moreover, KetoK is an unhydrolysable analogue of AcK that represents one of the most important posttranslationally modified amino acids in eukaryotic cells. Since protein acetylation is a reversible process catalyzed by enzymes like histone acetyltransferases and histone deacetyltransferases, direct incorporation of KetoK into proteins in vivo at sites with naturally occurring lysine acetylation will permanently install an AcK mimic and may provide an efficient way to decipher the regulation roles of acetylation in histones, p53 and other transcription regulatory proteins." The authors stated that the Methanomethylophilus alvus synthetase has not been previously used for KeK incorporation. While this is a technical extension (active sites transfer), it does not constitute a conceptual advance.

With due respect, the reviewer is merely spelling out our response to the respective comment of the reviewer in the first reviewing round (i.e. the reviewer cites the last 2 sentences of the discussion of the cited manuscript). We would like to emphasize that the mere notion that something could or should potentially be done in the future does - in our opinion - not suffice to conclude that a study providing evidence for such a hypothesis is not "novel" or does not provide a conceptual advance. We would also like to note that we never claimed that the use of the M. alvus system represents a conceptual advance.

2. I agree that the central claim is that KeK is a functionally reliable, non-hydrolyzable acetyllysine mimic to study protein acetylation in vitro and in cells. This is a broad claim that necessitates validation across a diverse range of biological processes. However, the authors fail to perform these key experiments in response to both reviewer #2 and #3. The study still relies heavily on structural similarity (NMR) and loss-of-function phenotypes. The current evidence, limited to only two proteins (Ub and p53), is not only insufficient in scope but also problematic in its interpretation, thereby failing to support this claim of generality.

1) In the Ubiquitin model: The primary functional readout (HDM2 autoubiquitination) only shows that KeK, like AcK, leads to a loss of function. This does not constitute positive evidence of functional mimicry. A true functional mimic should be able to recapitulate a gain of function or a specific switched function induced by acetylation, which is not shown here.

We agree with the reviewer that loss of function studies alone are not sufficient to conclude that KeK is a suitable mimic of AcK. This is why we have performed structural studies and more importantly interactome studies. Although we have no proof for the biological relevance of the

observed interactions (see also response to comment 4.2), the data clearly show that Ub11AcK interacts with a set of proteins that do not or not detectably interact with non-modified Ub (see also ref. 11 of the manuscript). That is, Ub11AcK has not just *lost* binding properties but has also *gained* binding properties. Importantly, the interaction properties of Ub11KeK are highly similar to Ub11AcK showing that in this case, KeK is a functional mimic of AcK.

2) My previous comment emphasized the necessity of evaluating the strategy's reliability and delving into deep biological questions. The authors' response misunderstands this point. I am not asking for a complete biological story on p53; rather, I am requesting evidence of the strategy's general applicability. This could be achieved by applying the KeK incorporation to a wider range of proteins or known acetylation-dependent processes, even at a preliminary level. Demonstrating that KeK can reliably mimic AcK in multiple, distinct systems would significantly strengthen the methodological foundation of the paper.

The reasons for studying p53 have been that we have a long-standing interest into p53 and that based on literature, we expected that it would be relatively straightforward to prove that KeK is a suitable mimic for AcK also in cells. The outcome is somewhat contrary to this expectation, mainly because the respective AcK variants were completely deacetylated in cells (and, thus, behaved like wild-type p53) and the KeK variants were inactive (based on literature, we would have expected that in some cases, the KeK variants are active). However, since we and others have previously shown that the respective AcK and KeK variants can still bind DNA (i.e. incorporation does not distort p53 structure), an important conclusion of our study is that results regarding the potential consequences of acetylation for p53 in cells that were obtained by indirect means need to be interpreted with caution.

We agree that it will be important to study the functionality of KeK incorporation on a more general level. However, based on our experience with p53, we feel that this needs to be done by groups with experience in characterizing the respective protein *in cells*. Moreover, we propose to rather use AcK for *in vitro* studies (as long as hydrolysis does not appear to be a matter of concern; see also our response to the comment of reviewer 2), as we have successfully done for additional proteins in bacteria (e.g. H1; ref. 47 of the manuscript), yet we are not in a position to functionally characterize these proteins in cells. We have added these important considerations to the Discussion (page 13, lines 345-349: *Because it is not hydrolyzable, KeK represents the AcK mimic of choice for such studies in cells, with the potential limitation that in some cases, protein(s) critically involved in executing the functional consequences of acetylation may recognize KeK less efficiently than AcK. For in vitro studies, we propose to use AcK, as long as hydrolysis is not a matter of concern.*)

3. HDM2 autoubiquitination quantification (Fig. 1b): A scientifically rigorous approach would involve using software like ImageJ to quantify the autoubiquitination level of HDM2 directly or consumption of free ubiquitin at different time points. This would allow for a direct comparison of ubiquitination kinetics between variants. The current reliance on visual inspection of Coomassie-stained gels is insufficient for a high-impact journal.

As requested, we quantified the levels of free ubiquitin. The relative levels are now indicated below the Coomassie-stained gel in Fig. 1b.

Figure presentation: I noticed the better image had been revised. However, the only adjustment of image brightness/contrast for the 11R lane, rather than ensuring equal protein loading across all samples, is not scientifically rigorous.

In this experiment, a master mix containing all the proteins required except the ubiquitin variants was employed to ensure that all the reactions contained the same amount of the individual proteins. Reactions were started by addition of the respective ubiquitin variant, the concentration of which was determined and normalized to the other Ub variants beforehand. Nonetheless, as we determined the levels of the free ubiquitin variants at $t = 0$ min, we can assure the reviewer that the amounts of the different ubiquitin variants employed were nearly identical as expected (with a variance of +/- 5 percent).

4. The authors correctly revised the high correlation and overlap between Ub 11AcK and Ub 11KeK interactomes. However, they dismiss the ~23% of differentially enriched proteins as likely technical artifacts from biological replicates, indirect interactome and MS methods. The conclusion remains further investigation.

1) A more thorough analysis should involve a functional enrichment analysis (e.g., GO, KEGG) of the proteins that are unique to Ub 11AcK or Ub 11KeK. Are these proteins enriched for specific pathways or functions? This could provide crucial clues about the subtle functional differences between AcK and its mimic, which is essential for the central claim that KeK is a functionally reliable, non-hydrolyzable acetyllysine mimic to study protein acetylation.

We had originally considered to perform a GO analysis. However, as the reviewer may be aware of, the involved numbers (i.e. 20 to 25% correspond to roughly 25 proteins in total) are too small for a meaningful GO analysis. We would also like to point out again that in the case of the Ub variants, KeK is as good of a functionally reliable AcK mimic as possible, as a difference of 20-25 percent in AE-MS experiments is not unusual, even with the "same" protein, when comparing biological replicates (e.g. Fernández-Costa et al. J Proteome Res 19, pp. 1697 (2020), PMID: 31880919; Poulos et al., Nat Commun 11, 3793 (2020), PMID: 32732981).

2) While cross-linking MS could offer higher confidence in identifying direct interactors, a compelling alternative to strengthen their findings would be to benchmark their identified interactors against previously published ubiquitin interactomes. Demonstrating that their AE-MS method recovers known, functionally relevant Ub-binding proteins would validate the reliability of their dataset and, by extension, the similarity between AcK and KeK.

There are basically only two ubiquitin interactomes available that may be comparable to the present one. The first one is from our group (ref. 11 of the manuscript) and the second from the group of M. Vermeulen (Zhang et al., Mol Cell 65, pp. 945 (2017); PMID: 28190767). However, the Vermeulen group used a different cell line, a different experimental setup, and other procedures for their analysis. Thus, the data sets are only comparable to a certain extent. Still, about 35 percent of the proteins found in our analysis were also found in the data set of the Vermeulen group. We also performed a GO analysis with our Ub data set. As expected, most of the proteins including a number of known Ub-binding proteins map to the keywords "proteasome" and "Ubl conjugation pathway". However, these data were discussed in our previous study. Thus, we decided not to include these in the present manuscript.

5. The authors argued that assessing the cytotoxicity of stable KeK incorporation is challenging. However, an initial evaluation below is both feasible and critical. Transient transfection of the KeK-incorporation system (including proper controls) coupled with a standard cell viability assay (e.g., MTT, CellTiter-Glo) would provide invaluable data. The goal is not to prove the method is non-toxic, but to transparently quantify its potential impact. Understanding the baseline cytotoxicity is

essential for future users to appropriately design their experiments and interpret results. A method's value is not diminished by its limitations, but by a failure to characterize them.

We routinely follow the viability of transfected cells by visual inspection and could never observe any signs of toxicity under the conditions used (transient transfection). That is, upon transfection the cells proliferate normally compared to a mock-transfected control. An example is shown in the figure for the reviewer, where we stained cells 60 h upon transfection with crystal violet (a common and fast approach to score for viable cells). We added a respective statement in the Methods section.

Figure for reviewer. H1299 cells were seeded in 6-well plates 24 h before transfection. Cells were transfected at an approximate confluency of 80% with Lipofectamine 2000 according to manufacturer's instructions with plasmids encoding wild-type p53 (wtp53), an expression construct for p53 harboring a TAG stop codon at position 120 (120TAG), and the AcK-RS/tRNA pair (RS) required for incorporation of KeK in different combinations as indicated in the absence or presence of KeK (final conc. indicated). 60 h after transfection, cells were confluent and there was no difference in viability, as indicated by crystal violet staining.

Response to reviewers' comments on "Non-hydrolyzable acetyllysine analogs to study protein acetylation in vitro and in cells" (NCOMMS-24-65817B)

Reviewer 2

I believe that with the qualifying statement the authors have added to the manuscript (p13), the overall conclusion is less likely to mislead readers. Personally, I would stop short of calling KeK the 'mimic of choice' for studies in cells (until there is broader functional validation), but it isn't my manuscript. I have no further suggestions to make.

In response to the reviewer's suggestion, we replaced "mimic/surrogate of choice" by "well-suited" in the Abstract (last sentence), Introduction (last paragraph, last sentence) and Discussion (page 13, second to last sentence).

Reviewer 3

The paper is ready to publish